# DyDiff: Long-Horizon Rollout via Dynamics Diffusion for Offline Reinforcement Learning

## Abstract

With the great success of diffusion models (DMs) in generating realistic synthetic vision data, many researchers have investigated their potential in decision-making and control. Most of these works utilized DMs to sample directly from the trajectory space, where DMs can be viewed as a combination of dynamics models and policies. In this work, we explore how to decouple DMs' ability as dynamics models in fully offline settings, allowing the learning policy to roll out trajectories. As DMs learn the data distribution from the dataset, their intrinsic policy is actually the behavior policy induced from the dataset, which results in a mismatch between the behavior policy and the learning policy. We propose Dynamics Diffusion, short as `DyDiff`, which can inject information from the learning policy to DMs iteratively. `DyDiff` ensures long-horizon rollout accuracy while maintaining policy consistency and can be easily deployed on model-free algorithms. We provide theoretical analysis to show the advantage of DMs on long-horizon rollout over models and demonstrate the effectiveness of `DyDiff` in the context of offline reinforcement learning, where the rollout dataset is provided but no online environment for interaction. Our code is at `https://anonymous.4open.science/r/DyDiff`.

## 1 Introduction

Diffusion models (DMs) have shown a remarkable ability to capture high-dimensional, multi-modal distributions and generate high-quality samples, such as images (Ho et al., 2020; Rombach et al., 2022), drug discovery (Xu et al., 2023), and motion generation (Tevet et al., 2022). Researchers find that such an ability also serves well in solving decision-making problems (Zhu et al., 2023b). For instance, using DMs as policy functions to generate single-step actions (Chi et al., 2023), as planners to generate trajectories guided by rewards or Q-functions (Janner et al., 2022; Zhu et al., 2023a), or as data synthesizers to learn the data distribution of the dataset and augment the dataset with more behavior data (He et al., 2024; Lu et al., 2024). Both diffusion planners and data synthesizers use DMs to generate long-horizon trajectories. However, they choose to directly sample from the trajectory space, resulting DMs a combination of dynamics models and policies, i.e., a policy (the dataset average policy or a high-rewarded policy) is embedded in the generated sequences. Thus, none of those DMs can serve as a dynamics model and generate trajectories for arbitrary policies.

In a preliminary study, we find that the ability to generate long-horizon rollouts can be much helpful in improving offline RL solutions. Specifically, we build a motivating example where a TD3BC (Fujimoto & Gu, 2021) agent is trained on an offline dataset with gradually augmenting on-policy data or dataset behavior data during learning, compared with no augmentation. Results in Fig. 1a reveal that *augmenting on-policy data is better than behavior data*. We further compare augmenting on-policy rollouts with different lengths, and the results plotted in Fig. 1b indicate that *augmenting long-horizon on-policy rollouts is better than shorter-horizon on-policy rollouts*.

Given the above findings, we hope to design a model that can synthesize long-horizon on-policy rollouts for offline policy training. In this paper, we propose a novel method named Dynamics Diffusion (`DyDiff`) to decouple existing DMs' roles as dynamics models and use their superior generative ability to accomplish this goal. Although some previous works have developed model-based methods for augmenting synthetic on-policy data via pre-trained single-step dynamics models (Yu et al., 2020; 2021), it is still difficult for them to generate long-horizon rollouts due to compounding errors. Different from them, `DyDiff` can model the interaction in the sequence level and generate long-horizon

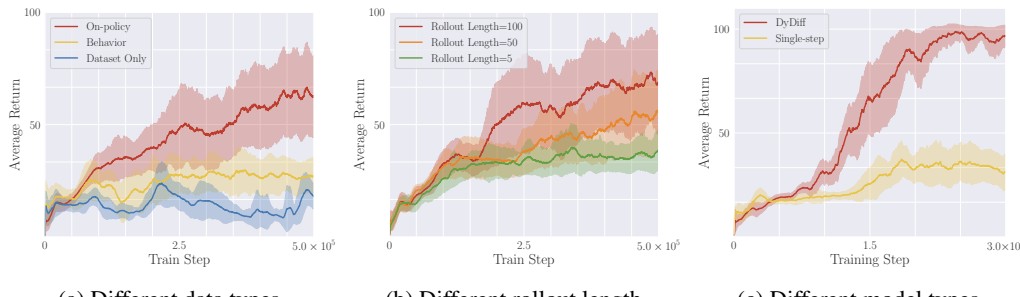

(a) Different data types.       (b) Different rollout length.       (c) Different model types.

Figure 1: Training the policy on a part of `hopper-medium-replay` dataset under different settings. **(a)** During training, we train a diffusion model to generate and gradually augment on-policy data and dataset behavior data, compared with no extra data augmented. **(b)** Augment model generated on-policy rollouts with different lengths. **(c)** Use single-step dynamics models and our `DyDiff` to generate rollouts. The detailed setting is described in Appendix A.

rollouts, which benefits the learning policy much more than shorter ones, as we showcase in Fig. 1b. The superiority of `DyDiff` in synthesizing long sequences over single-step models is also reflected in Fig. 1c, where the policy is augmented by rollouts with the same length but generated by `DyDiff` and single-step models, respectively.

To be more specific, `DyDiff` works by first running a pre-trained single-step dynamics model with the current policy for many steps to get the initial on-policy sequences; then, the trajectory served as the initial conditions for a diffusion model to generate new samples, which is further used for policy optimization. In this way, `DyDiff` combines the advantage of both the rollout consistency of single-step dynamics models with arbitrary policies, and the long-horizon generation of DMs with less compounding error. Theoretical analysis for `DyDiff` provides proofs of why DMs are better for long-horizon rollout than single-step dynamics model, and how the iterative process in `DyDiff` reduces the accumulated error of the synthetic trajectories.

We implement `DyDiff` as a plugin on a set of existing model-free algorithms, and conduct comprehensive experiments across various tasks on D4RL benchmarks, showing that `DyDiff` significantly improves the performance of these algorithms without any additional hyperparameter tuning.

In summary, our main contributions are listed as follows.

- **Investigating the policy mismatch problem**: We identify the policy mismatch problem in DMs for offline RL and investigate it in detail. To the best of our knowledge, this is the first work providing both theoretical and empirical analyses for this problem.
- **Developing the ability of DMs as dynamics models**: We propose a novel method named `DyDiff`, that combines DMs and single-step dynamics models, leveraging the advantages of both sides to perform long-horizon rollout with less compounding error.
- **Providing theoretical analysis for non-autoregressive generation**: We prove the advantage of non-autoregressive generation scheme against the autoregressive generation one in terms of the return gap between executing the policy in the real and the learned dynamics, where the former tightens the gap by a substantial factor of $\frac{\gamma}{1-\gamma} \frac{\epsilon_d}{\epsilon_m} \gg 1$.

## 2 RELATED WORK

**Diffusion Models in offline RL.** Diffusion models (Ho et al., 2020), a powerful class of generative models, have recently found applications in offline RL (Zhu et al., 2023b), serving as planners (Janner et al., 2022; Liang et al., 2023; He et al., 2024; Hu et al., 2023; Ajay et al., 2022; Zhu et al., 2023a) and policies (Wang et al., 2022; Chen et al., 2022; Lu et al., 2023; Hansen-Estruch et al., 2023; Kang et al., 2024). For instance, Diffusion QL (DiffQL) (Wang et al., 2022) employs a conditional diffusion model to represent the policy, aiming to maximize action-values during the training of the diffusion model. Additionally, Diffuser (Janner et al., 2022) proposes a novel data-driven decision-making approach based on trajectory-level diffusion probabilistic models. Recently, SynthER (Lu et al., 2024) utilizes diffusion models as data synthesizers for data augmentation in offline RL. The powerful

expressiveness of diffusion models enables non-autoregressive trajectory synthesis, which reduces compounding errors compared to multilayer perceptrons (MLPs). However, neglecting the learning policy results in a significant distribution gap between the generated data and the data sampled by the learning policy in the real environment, which hampers effective policy learning. In contrast, the proposed `DyDiff` leverages both the ability of non-autoregressive trajectory synthesis and information derived from the learning policy. A concurrent work, PGD (Jackson et al., 2024), also identifies the policy mismatch problem associated with diffusion models, but approaches it differently. It computes the log-likelihood of generated trajectories based on the learning policy, injecting this as guidance for diffusion models. However, their illustration is limited to toy environments. In this work, we investigate the policy mismatch issue from multiple perspectives, evaluating the algorithm in complicated locomotion tasks while providing a more comprehensive theoretical analysis.

**Offline model-based RL.** As an intersection of model-based RL and offline RL (Levine et al., 2020; Liu et al., 2021; Levine et al., 2020), offline model-based RL methods (Yu et al., 2021; 2020; Argenson & Dulac-Arnold, 2020; Kidambi et al., 2020; Matsushima et al., 2020; Swazinna et al., 2021) employ supervised learning and generative modeling techniques to improve policy performance. However, the distributional shift problem remains a fundamental challenge in offline model-based RL. On the one hand, many methods (Yu et al., 2020; 2021; Kidambi et al., 2020; Rigter et al., 2022; Li et al., 2024; Matsushima et al., 2020) adopt a conservative approach to utilizing the dynamics model, aiming to minimize estimation errors and enhance performance. For instance, MOPO (Yu et al., 2020) integrates uncertainty as a penalty term on the reward, while MOReL (Kidambi et al., 2020) estimates uncertainty by measuring the maximum discrepancy among ensemble models. This conservatism helps mitigate risks but may also limit the exploration of potentially beneficial actions. On the other hand, methods such as SynthER (Lu et al., 2024) leverage the dynamics model for data augmentation and successfully achieve high performance through enhanced data variety. Our approach takes into account information from the learning policy while intentionally avoiding overly conservative techniques, enabling the dynamics model to be fully leveraged without hindrance.

## 3 PRELIMINARIES

**Diffusion model.** Diffusion models (DMs) are a class of generative models that generate data $x_0$ by incrementally removing noise from a pure Gaussian distribution. In this work, we follow the architecture of EDM (Karras et al., 2022), which implements the forward process and the reverse process of the DM as the increase and decrease of the noise level of a probability flow ordinary differential equation (ODE) (Song et al., 2020b):

$$\mathrm{d}\boldsymbol{x} = -\dot{\sigma}(t)\sigma(t)\nabla_{\boldsymbol{x}} \log p(\boldsymbol{x}; \sigma(t))\mathrm{d}t \ , \tag{1}$$

where the dot denotes the derivative with respect to time. $\sigma(t)$ is the noise schedule with noise levels $\sigma^{\max} = \sigma^0 > \sigma^1 > \cdots > \sigma^N = 0$ . $\nabla_{\boldsymbol{x}} \log p(\boldsymbol{x}; \sigma(t))$ is the score function. We denote the data distribution at noise level $\sigma^i$ as $p(\boldsymbol{x}; \sigma^i)$ and the overall data distribution as $\sigma^{\mathrm{data}}$. In the forward process, noise is gradually added to the data $\boldsymbol{x}^N \sim p(\boldsymbol{x}; \sigma^N)$, transforming it into pure Gaussian noise. In contrast, during the reverse process, pure Gaussian noise is drawn from $\boldsymbol{x}^0 \sim p(\boldsymbol{x}; \sigma^0)$, and the sample is obtained by removing noise from $\boldsymbol{x}$. Please refer to Appendix B for more details.

**Offline RL.** Offline RL solves a Markov decision process (MDP) similar to online RL, but optimizes the policy solely using an offline dataset without interacting with the environment. Denote MDP $\mathcal{M} = \{\mathcal{S}, \mathcal{A}, T, r, \gamma, d_0\}$, where $\mathcal{S}, \mathcal{A}$ are the state space and the action space, $T(s'|s, a)$ is the dynamics function, $r(s, a)$ is the reward function, $\gamma \in (0, 1)$ is the discount factor, and $d_0$ is the initial state distribution. The formal objective of offline RL is to learn a policy $\pi$ that maximizes the discounted cumulative rewards as $\max_\pi J(\mathcal{M}, \pi) := \mathbb{E}_{s_0 \sim d_0, a_t \sim \pi(\cdot|s_t), s_{t+1} \sim T(\cdot|s_t, a_t)}[\sum_{t=0}^{\infty} \gamma^t r(s_t, a_t)] \ .$

## 4 DYNAMICS DIFFUSION (DYDIFF)

In this section, we present our design for generating synthetic data with DMs while ensuring consistency with the learning policy. We first detail the generation target of the DM and the sampling process. Next, we introduce the core of our method: how to use composite single-step dynamics models and DMs to generate data that adheres to the learning policy. Finally, we provide a theoretical analysis for our method, explaining why `DyDiff` outperforms the use of single-step models alone.

Figure 2: The sketch process of `DyDiff`. It mainly consists of three parts: (1) Sampling start states from $\mathcal{D}$ to generate initial trajectories as conditions with a single-step model. (2) Synthesizing rollout trajectories by iteratively sampling from the DM and the learning policy. (3) Filtering synthesized data and adding high-reward trajectories to $\mathcal{D}_{\text{syn}}$.

The sketch process of `DyDiff` is illustrated in Fig. 2. Generally, `DyDiff` begins by sampling states from the real dataset $\mathcal{D}$ as initial states of rollout, and an action sequence is derived from interaction between a single-step dynamics model and the learning policy for each initial state. A DM, conditioned on the initial state and the action sequence, is then employed to synthesize the corresponding state sequence. This state sequence is iteratively refined using both the learning policy and the DM. Finally, a reward-based filter is applied to select high-reward data, which are added to the synthetic dataset $\mathcal{D}_{\text{syn}}$ for further policy training.

### 4.1 DIFFUSION MODELS AS ROLLOUT SYNTHESIZER

DMs demonstrate a remarkable ability to model complex distributions and have been utilized for synthesizing sequential data in offline RL in many previous works (Ajay et al., 2022; Zhu et al., 2023a; Lu et al., 2024). Since offline RL possesses a pre-collected dataset $\mathcal{D}$ containing trajectory-level sequential data, we can easily pre-train DMs over $\mathcal{D}$ via supervised learning. We first construct the training set for the DM from $\mathcal{D}$. Let $L$ denote the length of the generation sequence for the DM. For a trajectory $\tau = (s_0, a_0, s_1, \ldots, a_{H-1}, s_H) \in \mathcal{D}$, the corresponding training trajectories are derived by slicing or padding $\tau$ to a length of $L$, i.e. containing $L+1$ states and $L$ actions:

$$\mathcal{S}(\tau) = \begin{cases} \{\tilde{\tau}_i = (s_i, a_i, s_{i+1}, a_{i+1}, \ldots, a_{i+L-1}, s_{i+L}) \mid 0 \leq i \leq H - L\} & (H \geq L) \\ \{\tilde{\tau} = (s_0, a_0, s_1, \ldots, a_{H-1}, s_H, 0, 0, \ldots, 0) \mid |\tilde{\tau}| = L\} & (H < L) \end{cases}. \quad (2)$$

The training set for the DM is the union of $\mathcal{S}(\tau)$ over all trajectories in $\mathcal{D}$, defined as $\mathcal{S} = \bigcup_{\tau \in \mathcal{D}} \mathcal{S}(\tau)$. Without causing ambiguity, we will also denote the trajectory in $\mathcal{S}$ as $\tau$ for simplicity.

There are several possible choices regarding which part of the trajectories the DM will generate. DecisionDiffuser (Ajay et al., 2022) generates state sequences, MTDiff (He et al., 2024) generates state-action sequences, whereas SynthER (Lu et al., 2024) generates state-action-reward sequences. To leave room for the learning policy, we generate only the state sequence $\tau_s = (s_0, s_1, \ldots, s_L)$ of a trajectory $\tau = (\tau_s, \tau_a)$, conditioned on the action part $\tau_a = (a_0, a_1, \ldots, a_{L-1})$ and the initial state $s_0$. Empirically, we generate both states and actions simultaneously, but replace the generated actions and the initial state with the given conditions after each diffusion step. This scheme effectively injects the conditions into the diffusion process, while preserving the relative positions between states and actions, enabling the DM to learn their causal relation. Formally, suppose the DM produces $\tau^i$ after the $i$-th denoising step. The conditions are applied by a hard replacement as

$$\tau^i = (s_0^i, a_0^i, s_1^i, a_1^i, s_2^i, \ldots, a_{L-1}^i, s_L^i) \xrightarrow{\text{Apply Conditions}} \tau^i = (s_0, a_0, s_1^i, a_1, s_2^i, \ldots, a_{L-1}, s_L^i). \quad (3)$$

We follow EDM (Karras et al., 2022) to train and sample from the DM, which uses a neural network $D_\theta$ to directly predict the denoised sample from the noisy one, instead of predicting the noise. Let

$\hat{\tau}^N = D_\theta(\tau^i)$ be the predicted denoised trajectory from $\tau^i$. Denote $\hat{\tau}^N_{s>0} = (\hat{s}^N_1, \hat{s}^N_2, \ldots, \hat{s}^N_L)$ for the predicted state sequence and $\hat{\tau}^N_a = (\hat{a}^N_0, \hat{a}^N_1, \ldots, \hat{a}^N_{L-1})$ for the action sequence. With hard-replaced conditions, $\hat{\tau}^N_a$ always equals the given condition $\tau_a$, and $\hat{s}^N_0$ equals $s_0$. Therefore, we only need to compute the loss between $\hat{\tau}^N_{s>0}$ and $\tau_{s>0}$. The overall training loss for $D_\theta$ is

$$L_{\text{diff}}(\theta) = \mathbb{E}_{\tau \sim \mathcal{S}, \sigma \sim p_\sigma, \boldsymbol{n} \sim \mathcal{N}(0, \sigma^2 \boldsymbol{I})}[\lambda(\sigma)\|\tau^N_{s>0} - \tau_{s>0}\|^2_2], \text{ where } (s^N_0, \tau^N_{s>0}, \tau^N_a) = D_\theta(\tau + \boldsymbol{n}; \sigma) . \tag{4}$$

Here, $\sigma$ is the noise scale, $p_\sigma$ is the distribution of $\sigma$, and $\lambda(\sigma)$ gives weights for different noise scales. We follow the same configuration as EDM, with detailed values listed in Appendix C. Under Eq. (4), we expect the DM to learn the environment dynamics from the dataset.

With a trained DM $D_\theta$, we can now sample a state sequence $\tau^N_s$ beginning from $s_0$ and corresponding to a given action sequence $\tau_a$, starting from pure noise $\tau^0 \sim \mathcal{N}(0, t^2_0 \boldsymbol{I})$. We utilize the EDM sampler for improved sampling accuracy and speed, with a slightly modification in the denoising part to apply the conditions. Most of the sampling process remains identical to EDM, so we provide the details in Algo. 2 in Appendix C. For brevity, we denote this sampling process as drawing from the distribution $p_\theta(\tau|s_0, \tau_a)$.

Though we can now use DMs to generate state trajectories, the choice of initial action trajectory is worth considering. Relying on random action trajectories would produce low-reward samples, as it is equivalent to executing a random policy from $s_0$. Moreover, directly picking an real action sequence from the dataset would still correspond to the underlying behavior policy rather than the learning policy, which fails to meet our goal of maintaining policy consistency. Therefore, we need to derive the initial action trajectory with the assistance of a single-step dynamics model. Besides, we do not incorporate policy information in the generation process of the DM, so the immediate synthetic trajectories requires further refinement. We will introduce the details in the next section.

### 4.2 REFINE ROLLOUTS WITH DIFFUSION MODELS

To obtain a good initial action sequence, we allow the learning policy to interact with a pre-trained single-step dynamics model $T_\phi(s, a)$ parameterized by $\phi$. This model is directly trained via supervised learning over the dataset $\mathcal{D}$, with the following loss objective:

$$L_{\text{dyn}}(\phi) = \mathbb{E}_{(s,a,s') \sim \mathcal{D}, \hat{s}' \sim T_\phi(s,a)}[\|\hat{s}' - s'\|^2_2] . \tag{5}$$

For interaction, the most straightforward approach is to start from an initial state $s_0$ sampled from $\mathcal{D}$, and sample $\hat{a}_0$ from the learning policy $\pi(\cdot|s_0)$. The dynamics model then predicts the next state $\hat{s}_1 \sim T_\phi(\cdot|s_0, \hat{a}_1)$. By iteratively sampling from the policy and the dynamics model, we can form a rollout trajectory autoregressively as

$$\hat{\tau}_{\text{dyn}} = (s_0, \hat{a}_0, \hat{s}_1, \ldots, \hat{a}_{L-1}, \hat{s}_L), \quad \hat{a}_i \sim \pi(\cdot|\hat{s}_i), \hat{s}_{i+1} \sim T_\phi(\cdot|\hat{s}_i, \hat{a}_i), 0 \le i \le L - 1 , \tag{6}$$

where $L$ is the rollout length and $\hat{s}_0 := s_0$. However, rollout by interacting with a single-step dynamics model leads to severe compounding error as $L$ increases, thus not benefiting policy training as shown in Fig. 1c. Therefore, $\hat{\tau}_{\text{dyn}}$ is not directly used for policy improvement but only as an initial condition for the DM, which can generate more accurate trajectories. As all actions of $\hat{\tau}_{\text{dyn}}$ are sampled from the learning policy $\pi$, $\hat{\tau}_{\text{dyn}}$ naturally ensures policy consistency, making it a suitable initial condition for $p_\theta$. Formally, we select the action sequence $\hat{\tau}_{a,\text{dyn}}$ and the first state $s_0$ as conditions, sampling a new trajectory from $p_\theta(\tau|s_0, \tau_a)$:

$$(s_0, \hat{\tau}^{(1)}_{s,\text{DM}}, \hat{\tau}_{a,\text{dyn}}) \sim p_\theta(\cdot|s_0, \hat{\tau}_{a,\text{dyn}}) . \tag{7}$$

Here, we use $\hat{\tau}^{(k)}_{\text{DM}}$ to represent the synthetic trajectory after the $k$-th generation. However, the diffusion sampling process only modifies the state sequence while preserving $s_0$ and $\hat{\tau}_{a,\text{dyn}}$ unchanged, which violates the policy consistency. To correct this, we resample the action sequence from the learning policy given $s_0$ and $\hat{\tau}^{(1)}_{s,\text{DM}}$:

$$\hat{a}^{(1)}_{0,\text{DM}} \sim \pi(\cdot|s_0), \quad \hat{a}^{(1)}_{i,\text{DM}} \sim \pi(\cdot|\hat{s}^{(1)}_{i,\text{DM}}), \quad \text{where } 1 \le i \le L - 1 . \tag{8}$$

Now, $\hat{\tau}^{(1)}_{\text{DM}} = (s_0, \hat{\tau}^{(1)}_{s,\text{DM}}, \hat{\tau}^{(1)}_{a,\text{DM}})$ is consistent with the learning policy but violates the dynamics. We address this in the same way as $\hat{\tau}_{\text{dyn}}$, by sampling a new trajectory from the DM $p_\theta$ given $s_0$ and

$\hat{\tau}_{a,\text{DM}}^{(1)}$ as conditions:

$$(s_0, \hat{\tau}_{s,\text{DM}}^{(2)}, \hat{\tau}_{a,\text{DM}}^{(1)}) \sim p_\theta(\cdot | s_0, \hat{\tau}_{a,\text{DM}}^{(1)}) . \tag{9}$$

Then, the learning policy $\pi$ is used to correct the action sequence, ensuring policy consistency. By iteratively applying the DM and the learning policy, we can gradually inject information about the learning policy into the generated trajectory while maintaining the dynamics accuracy with the DM.

Finally, we denote the final trajectory after $M$ iterations as $(s_0, \hat{\tau}_{a,\text{DM}}, \hat{\tau}_{s,\text{DM}}) = \hat{\tau}_{\text{DM}} := \hat{\tau}_{\text{DM}}^{(M)}$. Following the scheme of MBPO (Janner et al., 2019), we create another replay buffer $\mathcal{D}_{\text{syn}}$ to store synthetic data. In practice, a batch of states is uniformly sampled from the real dataset $\mathcal{D}$ as initial states, denoted as $\mathcal{B}_s = \{(s_0)_k\}_{k=1}^{B_r}$, where $B_r$ is the batch size of the rollout. Each initial state $s_0$ will induce a rollout trajectory $\hat{\tau}_{\text{DM}}$, so $\mathcal{B}_s$ derives a trajectory set $\mathcal{B}_\tau = \{(\hat{\tau}_{\text{DM}})_k\}_{k=1}^{B_r}$. To prevent data with low rewards from negatively impacting policy training, we filter $\mathcal{B}_\tau$ using a reward-based filter before adding the rollout trajectories into $\mathcal{D}_{\text{syn}}$. As we do not have direct access to the actual reward function, we pre-train a reward model $r_\psi(s, a)$ that predicts the rewards of synthetic transitions. Similar to the dynamics model, $r_\psi$ is simply trained through supervised learning:

$$L_{\text{rew}}(\psi) = \mathbb{E}_{(s,a,r)\sim\mathcal{D}, \hat{r}\sim r_\psi(s,a)}[(\hat{r} - r)^2] . \tag{10}$$

For filtering, we predict the reward for each transition in $\hat{\tau}_{\text{DM}}$ and sum them up for the entire trajectory:

$$r_\psi(\hat{\tau}_{\text{DM}}) := r_\psi(s_0, \hat{a}_{0,\text{DM}}) + \sum_{i=1}^{L-1} r_\psi(\hat{s}_{i,\text{DM}}, \hat{a}_{i,\text{DM}}) . \tag{11}$$

Only a proportion $\eta$ of trajectories in $\mathcal{B}_\tau$ is added to $\mathcal{D}_{\text{syn}}$. We introduce two filtering schemes to select high-reward data as follows:

- **Hardmax**: Sort the trajectories by their accumulative rewards and directly select $\lfloor \eta B_r \rfloor$ of them with the highest rewards.

- **Softmax**: Calculate a probability distribution $p_r((\hat{\tau}_{\text{DM}})_k) = \frac{\exp(r_\psi((\hat{\tau}_{\text{DM}})_k))}{\sum_{j=1}^{B_r} \exp(r_\psi((\hat{\tau}_{\text{DM}})_j))}$ using the softmax of their accumulative rewards, and sample $\lfloor \eta B_r \rfloor$ of them according to $p_r$.

Intuitively, the hardmax filter strictly selects trajectories with high rewards, while the softmax filter includes those with low rewards. However, considering that offline RL policies can outperform the behavior policy by stitching together trajectories in the dataset, the softmax filter provides greater diversity and opportunities for the policy to discover better patterns.

As `DyDiff` is an add-on scheme for synthesizing data, we do not design additional policy training algorithms but instead directly incorporate existing model-free offline policy training methods that explicitly require policies. Our overall algorithm is summarized in Algo. 1.

### 4.3 THEORETICAL ANALYSIS

We provide a brief theoretical analysis to show why models supporting non-autoregressive generation, such as DMs, are superior than single-step models. The following analysis is  Let $T(s'|s, a)$ be the real dynamics function. We begin with a lemma from MBPO (Janner et al., 2019) that bounds the return gap between the real dynamics and the learned single-step dynamics. Denote the accumulative discounted return in dynamics $T$ with policy $\pi$ as $J(T, \pi)$, and the maximum reward as $R$.

**Lemma 1.** *(Lemma B.3 of MBPO). Suppose the error of a single-step dynamics model $T_m(s'|s, a)$ can be bounded as $\max_t \mathbb{E}_{a\sim\pi}[D_{\text{KL}}(T_m(s'|s, a)\|T(s'|s, a))] \leq \epsilon_m$. Then after executing the same policy $\pi$ from the same initial state $s_0$ in $T_m$ and the real dynamics $T$, the expected returns are bounded as*

$$|J(T, \pi) - J(T_m, \pi)| \leq \frac{2R\gamma\epsilon_m}{(1-\gamma)^2} . \tag{12}$$

Note that this formulation differs slightly from its original version in MBPO, as there is no policy error term; the policies executed in both the trained dynamics model and the real dynamics are the same in offline RL. Then, the return gap of DMs can also be bounded. Denote the state distribution after executing an action sequence $\tau_a$ from $s_0$ in the real dynamics as $T(s_t|s_0, \tau_a)$, and the state distribution induced by the DM conditioned on $s_0$ and $\tau_a$ as $T_d(s_t|s_0, \tau_a)$.

**Theorem 1.** *Suppose the error of a non-autoregressive model $T_d(s_t|s_0, \tau_a)$ can be bounded as $\max_t D_{\text{TV}}(T_d(s_t|s_0, \tau_a))\|T(s_t|s_0, \tau_a) \leq \epsilon_d$. Then after executing the same policy $\pi$ from the same initial state $s_0$ in $T_d$ and the real dynamics $T$, the expected returns are bounded as*

$$|J(T, \pi) - J(T_d, \pi)| \leq \frac{2R\epsilon_d}{1 - \gamma} \ . \tag{13}$$

The proof is provided in the Appendix D. We observe that these two bounds differ by a multiplier $\frac{\gamma}{1-\gamma}\frac{\epsilon_m}{\epsilon_d}$. The first part, $\frac{\gamma}{1-\gamma}$, is greater than 1 when $0.5 < \gamma < 1$. In practice, $\gamma$ is typically set above 0.9. For the second part, although $\epsilon_m$ bounds the single-step error and $\epsilon_d$ bounds the accumulative multi-step error, we still have $\epsilon_d \approx \epsilon_m$ due to the superior modeling capabilities of DMs. Consequently, the inequality $\frac{\gamma}{1-\gamma}\frac{\epsilon_m}{\epsilon_d} > 1$ holds, indicating that the non-autoregressive models enjoy a better return gap than single-step models. The difference in the multiplier arises from the fact that the non-autoregressive model is merely affected by the compounding error. However, both $\epsilon_d$ and $\epsilon_m$ are related to complicated neural networks without theoretical analysis so far, they cannot be further decomposed analytically. To validate our assumptions on the error rates of single-step models versus DMs, we conduct a simple experiment to compute the MSE of rollouts generated by both models. The results support that $\epsilon_d < \epsilon_m$ over long horizons. Detailed settings and results are provided in Appendix D.3. Finally, we would like to clarify that the theoretical analysis applies to general non-autoregressive models, with DMs and `DyDiff` serving as specific examples. It highlights the potential of using non-autoregressive models for synthesizing rollouts.

Next, we analyze the effect of the iteration times $M$. In `DyDiff`, we start from the state trajectory generated by the autoregressive model, and iterate between the DM and the learning policy for $M$ times. While non-autoregressive models demonstrate greater accuracy than single-step models at the transition level, their performance at the trajectory level warrants further investigation. Let $\tau = (s_0, a_0, s_1, \ldots) = (\tau_s, \tau_a)$ denote the trajectory from $s_0$ induced by $\pi$ in the real dynamics. We define $\tau_m = (s_0, a_{0,m}, s_{1,m}, \ldots) = (\tau_{s,m}, \tau_{a,m})$ as the trajectory generated autoregressively, and $\tau_d^{(k)} = (s_0, a_{0,d}^{(k)}, s_{1,d}^{(k)}, a_{1,d}^{(k)}, s_{2,d}^{(k)}, \ldots) = (\tau_{s,d}^{(k)}, \tau_{a,d}^{(k)})$ as generated non-autoregressively after the $k$-th iteration. We begin with assumptions on the state distribution distance between $\tau_s$ and $\tau_{s,d}$ under different action sequences.

**Assumption 1.** *The error between $T(s_t|s_0, \tau_a)$ and $T_d(s_t|s_0, \tau_{a,d})$ can be bounded as $\max_t D_{\text{TV}}(T_d(s_t|s_0, \tau_{a,d})\|T(s_t|s_0, \tau_a)) \leq \epsilon_{s,d} + C_{a,d}\max_t \|\tau_{a,d} - \tau_a\|$, where $C_{a,d}$ is a constant.*

**Assumption 2.** *Given two state sequences $\tau_{s,1}$ and $\tau_{s,2}$, the distance between corresponding action sequences induced by $\pi$ is bounded as $\max_t D_{\text{TV}}(\pi(\tau_a|\tau_{s,1})\|\pi(\tau_a|\tau_{s,2})) \leq C_\pi \max_t \|\tau_{s,1} - \tau_{s,2}\|$, where $C_\pi$ is a constant.*

Assumption 1 is very similar to the condition outlined in Theorem 1, but it also takes into account the difference in the action sequences. Intuitively, the error of the non-autoregressive model is distributed across the entire trajectory, which suggests the change in the action sequence will not result in significant differences in the state sequence. Assumption 2 reflects the smoothness of the policy. Now, we derive how the distance between $\tau_{s,d}^{(k)}$ and $\tau_s$ evolves over iterations. The error of the initial state sequence $\tau_{s,m}$ is given by Lemma 2 in Appendix D, specifically $L\epsilon_m$. Then, the error of the initial action sequence is

$$d(\tau_{a,m}, \tau_a) = \max_t D_{\text{TV}}(\pi(\tau_a|\tau_{s,m})\|\pi(\tau_a|\tau_s)) \leq C_\pi L\epsilon_m \ . \tag{14}$$

We then sample a new state trajectory $\tau_{s,d}^{(1)}$ from $p_\theta(\tau|s_0, \tau_{a,m})$. Under Assumption 1, the error of $\tau_{s,d}^{(1)}$ is bounded as

$$d(\tau_{s,d}^{(1)}, \tau_s) = \max_t D_{\text{TV}}(T_d(s_t|\tau_{a,m}, s_0)\|T(s_t|\tau_a, s_0)) \leq \epsilon_{s,d} + C_{a,d}C_\pi L\epsilon_m \ . \tag{15}$$

This state sequence is then fed into the policy $\pi$ to compute the corresponding action sequence $\tau_{a,d}^{(1)}$, and its error is bounded as

$$d(\tau_{a,d}^{(1)}, \tau_a) = \max_t D_{\text{TV}}(\pi(\tau_a|\tau_{s,d}^{(1)})\|\pi(\tau_a|\tau_s)) \leq C_\pi(\epsilon_{s,d} + C_{a,d}C_\pi L\epsilon_m) \ . \tag{16}$$

From Eq. (15) and Eq. (16), each iteration introduces both additive and multiplicative constant coefficients to the error bound. Continuing the iterations, we can derive the error of the state sequence

after the $k$-th iteration as

$$d(\tau_{s,d}^{(k)}, \tau_s) = \max_t D_{\text{TV}}(T_d(s_t|\tau_{a,d}^{(k-1)}, s_0)\|T(s_t|\tau_a, s_0)) \leq \frac{1-C^k}{1-C}\epsilon_{s,d} + C^k L\epsilon_m, \quad k = 1, 2, \ldots, \quad (17)$$

where $C = C_{a,d}C_\pi$. As $k$ increases, the error bound evolves from $L\epsilon_m$ to $\epsilon_{s,d}/(1-C)$. In practice, the accuracy of DMs is generally much better than that of auto-regressive models, which implies $\epsilon_{s,d} \ll L\epsilon_m$. This shows that the iterating optimizes the error bound of the synthetic trajectory.

Finally, it is important to note that increasing the iteration times $M$ will not necessarily lead to improved performance. Too many iterations may push the intermediate result out of the dataset's coverage, reducing the accuracy of the DM. Additionally, large $M$ can significantly increase rollout time, as each rollout requires sampling from the DM $M$ times. Therefore, the choice of $M$ should be determined based on the complexity of the dataset and the structure of the DM. Further discussions can be found in Section 5.4.

## 5 EXPERIMENTS

To validate the effectiveness and generalization capability of `DyDiff`, we conduct extensive experiments across various benchmark tasks and different offline model-free policy training algorithms. Our experiments are designed to answer the following key research questions:

- Can `DyDiff` effectively enhance the performance of underlying policies without requiring policy hyperparameter tuning?

- Is `DyDiff` adaptable to different types of tasks, including dense- and sparse-reward tasks?

- How do different critical hyperparameters impact the performance of `DyDiff`?

### 5.1 EXPERIMENT SETTINGS

We conduct the experiments on the D4RL (Fu et al., 2020) offline benchmark, following the common standards as previous offline RL studies. Specifically, we evaluated our performance on MuJoCo locomotion tasks and Maze2d, with the former characterized as dense-reward tasks and the latter as sparse-reward tasks. For each MuJoCo locomotion task, three datasets are included: (a) `medium-replay`, shorted as `mr`, containing data collected by a policy during its online training process, ranging from stochastic to medium-level. (b) `medium`, shorted as `md`, containing data collected by a single medium-level policy. (c) `medium-expert`, shorted as `me`, containing a 50/50 mixture of data collected by a medium policy and an expert policy, respectively. In summary, `mr` and `me` are mixed dataset, while `md` is a single-policy datasets. For Maze2d, we evaluated all three difficulties: umaze, medium, and large, from easy to hard. The harder the task, the larger and more intricate the maze becomes.

For the underlying policy, we select three popular state-of-the-art offline RL algorithms: CQL (Kumar et al., 2020), TD3BC (Fujimoto & Gu, 2021), and DiffQL (Wang et al., 2022). CQL is a Q-constraint method that employs a stochastic Gaussian policy, while TD3BC is a straightforward modification of TD3 (Fujimoto et al., 2018) using a deterministic policy. DiffQL is a recent Q-learning method that incorporates DMs as policies. Our choices for baseline cover various types of the learning policy. Note that we omit IQL (Kostrikov et al., 2021) as our underlying policy, since it only trains the value and Q-functions without an explicit policy, which does not align with our goal of reducing the gap to the learning policy. All underlying policies are reimplemented in our codebase for fair comparison. We test both hardmax and softmax filters and report the results of the softmax filter here. The full results are detailed in Appendix E.2.

In addition to the underlying policies as baselines, we also compare `DyDiff` to SynthER (Lu et al., 2024), an add-on data augmentation method that utilizes DMs to synthesize trajectories. SynthER is similarly reimplemented and added on the same base policies.

### 5.2 RESULTS

The main results for D4RL MuJoCo locomotion tasks are presented in Tab. 1, demonstrating that `DyDiff` improves base policies across most datasets, and achieving comparable performance in the

Table 1: Results on MuJoCo locomotion tasks. The reported number is the normalized score, averaged over 3 seeds and last 5 epochs, $\pm$ standard deviation. Note that our method is an add-on method to model-free offline algorithms, we reimplement the baselines in the same codebase of `DyDiff` for fair comparison. The best average results are in **bold**.

| Dataset | TD3BC | | | CQL | | | DiffQL | | |
|---|---|---|---|---|---|---|---|---|---|
| | Base | SynthER | **DyDiff** | Base | SynthER | **DyDiff** | Base | SynthER | **DyDiff** |
| hopper-md | 65.8±5.8 | 59.0±5.2 | 71.5±15.5 | 57.9±3.7 | 57.1±2.3 | 54.9±2.3 | 60.2±3.6 | 58.9±2.9 | 55.1±2.6 |
| hopper-me | 95.2±14.9 | 94.1±12.3 | 98.4±13.4 | 85.3±9.8 | 92.3±7.4 | 90.9±8.2 | 109.0±4.6 | 108.2±4.8 | 109.1±3.7 |
| hopper-mr | 81.5±17.4 | 50.4±13.4 | 82.6±20.1 | 87.7±7.8 | 92.4±6.5 | 95.3±2.6 | 97.8±5.1 | 99.1±4.4 | 99.5±3.4 |
| halfcheetah-md | 50.6±0.5 | 51.2±2.9 | 58.9±2.1 | 43.8±2.6 | 43.7±0.2 | 43.2±1.1 | 47.1±2.5 | 47.3±2.6 | 54.9±4.6 |
| halfcheetah-me | 69.7±18.4 | 80.0±7.5 | 77.6±10.6 | 53.0±9.0 | 49.4±5.1 | 60.8±9.2 | 94.1±0.7 | 90.2±4.7 | 94.5±2.0 |
| halfcheetah-mr | 46.0±0.6 | 45.2±0.4 | 44.2±6.1 | 42.9±2.6 | 43.2±0.3 | 41.5±2.2 | 45.1±4.1 | 46.0±2.8 | 47.5±5.7 |
| walker2d-md | 76.8±16.3 | 83.5±2.1 | 87.9±1.1 | 79.3±2.4 | 82.5±1.1 | 79.4±0.2 | 84.3±0.8 | 85.0±1.3 | 83.3±1.9 |
| walker2d-me | 110.7±0.6 | 110.6±0.4 | 110.6±1.3 | 108.9±0.6 | 109.1±0.4 | 108.8±0.4 | 109.6±0.2 | 109.8±0.4 | 109.7±0.3 |
| walker2d-mr | 85.8±11.8 | 90.4±5.3 | 74.5±8.9 | 80.5±3.7 | 85.7±2.8 | 86.8±7.0 | 90.6±1.9 | 94.4±3.5 | 92.3±2.2 |
| **Average** | 75.8 | 73.8 | **79.6** | 71.0 | 72.8 | **73.5** | 82.0 | 82.1 | **82.9** |

remaining ones. Our reimplemented baselines yield similar performance compared to their original papers, except SynthER, which enlarges the size of the base policy networks, a change we do not implement in our reimplementation. Moreover, we maintain the original hyperparameters of all base algorithms. Detailed settings and hyperparameters are described in Appendix E.

Among the various datasets (`md`, `me`, and `mr`), `DyDiff` performs well on `mr` and `me` datasets but fails to improve the baselines on `md`. A possible reason is that the data coverage of `md` is so narrow that the intermediate results of the sampling iterations fall out of distribution, leading to a decrease in data accuracy. In contrast, `DyDiff` effectively generates high-quality, diversified data when the data coverage is broad, thereby enhancing the base policies. Furthermore, as the synthetic data aligns with the distribution of the learning policy, it promotes better performance than SynthER, which uniformly upsamples the entire dataset. From the perspective of different base policies, `DyDiff` exhibits relative incompatibility with CQL. The computation of the conservative term in CQL relies on Q-values on out-of-distribution data, making CQL more sensitive to data accuracy.

## 5.3 Experiments on Sparse-reward Tasks

Table 2: Results on Maze2d tasks. We report average normalized scores over 3 independent runs, $\pm$ standard deviation. The best average results are in **bold**.

| Dataset | TD3BC | | | CQL | | | DiffQL | | |
|---|---|---|---|---|---|---|---|---|---|
| | Base | SynthER | **DyDiff** | Base | SynthER | **DyDiff** | Base | SynthER | **DyDiff** |
| maze2d-umaze | 0.35±0.10 | 0.32±0.09 | 0.55±0.12 | 0.19±0.15 | 0.10±0.12 | 0.58±0.43 | 0.47±0.01 | 0.45±0.02 | 0.46±0.02 |
| maze2d-medium | 0.81±0.50 | 0.49±0.20 | 1.34±0.19 | 0.93±0.13 | 0.92±0.03 | 1.56±0.17 | 0.50±0.02 | 0.17±0.04 | 1.62±0.02 |
| maze2d-large | 0.43±0.46 | 0.98±0.33 | 1.82±0.42 | 0.05±0.11 | 0.37±0.05 | 1.10±0.07 | 1.09±0.29 | 1.38±0.26 | 1.97±0.15 |
| Average | 0.53 | 0.60 | **1.24** | 0.39 | 0.46 | **1.08** | 0.69 | 0.67 | **1.35** |

For sparse-reward environments, we evaluate `DyDiff` across Maze2d tasks of varying difficulties, as presented in Tab. 2. It shows that `DyDiff` consistently improves the base policy, particularly in the more challenging `maze2d-medium` and `maze2d-large` tasks. In these environments, the agent only receives rewards when approaching the goal, leaving most transitions in the offline dataset with zero reward. Consequently, the policy training algorithm must "stitch" together partial trajectories to discover the optimal path to the goal. This stitching process is highly challenging due to the sparse reward signal. However, `DyDiff` alleviates this difficulty by leveraging its ability to generate long-horizon trajectories. By synthesizing full trajectories that guide the agent directly toward the goal, `DyDiff` reduces the reliance on stitching partial trajectories, thereby accelerating learning and improving policy performance. In contrast, SynthER, which merely upsamples the dataset uniformly, lacks the capability to integrate long-horizon information meaningfully, thus offering less assistance during policy training.

## 5.4 Ablation Studies

To verify our theoretical analysis and assess the sensitivity of `DyDiff` to key hyperparameters, we conduct experiments on varying the iteration times $M$, rollout length $L$, filter proportion $\eta$, and real

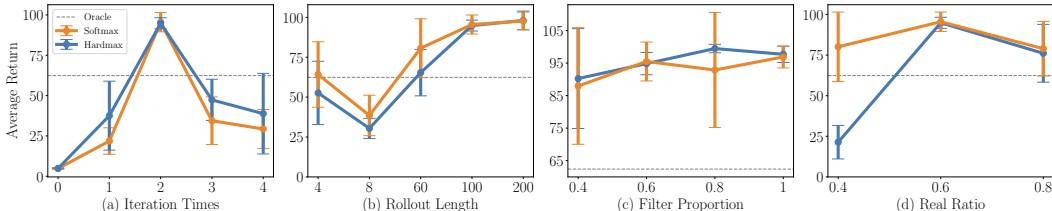

Figure 3: Ablation studies on various hyperparameters. Experiments on iteration times and rollout length validate our theory analysis, whereas those on filter proportion and real ratio prove the robustness of `DyDiff`.

ratio $\alpha$. The former three hyperparameters have been introduced above, and the last real ratio $\alpha$ is commonly used in MBRL to control the proportion of the real data used in policy training (Lai et al., 2021). All ablation studies are performed on the `hopper-mr` dataset using the TD3BC base policy.

**Iteration time.** As discussed in Section 4.3, larger iteration times reduces the error bound but increases the probability of falling out of the data distribution, which may degrade the data accuracy. Fig. 3a proves our analysis that a medium $M$ yields the best performance. Note that when $M = 0$, `DyDiff` reverts to only using single-step models for rollout. This also highlights the ability of DMs on long-horizon generation against single-step models.

**Rollout length.** As illustrated in Fig. 1b, large rollout length benefits the exploration of the policy. However, longer rollouts also increase $\epsilon_d$, loosening the return gap. We test `DyDiff` across various rollout lengths, with results presented in Fig. 3b. These results support our analysis of $L$, showcasing that DMs have a greater potential than single-step models due to their ability to generate accurate long-horizon trajectories.

**Filter proportion.** This hyperparameter controls the amount of data added to $\mathcal{D}_{\text{syn}}$ during each rollout. Intuitively, a higher $\eta$ increases the data diversity but may also introduce more low-reward data, and vice versa. The results in Fig. 3c show that `DyDiff` is robust in $\eta$, suggesting the high quality of generated data.

**Real ratio.** The real ratio determines the proportion of the real data when sampling from $\mathcal{D}$ and $\mathcal{D}_{\text{syn}}$. Since `DyDiff` only does rollout from real initial states, it is not feasible to entirely replace the real data with synthetic data as SynthER. We begin with a commonly used setting of $\alpha = 0.6$ and evaluate different $\alpha$. The results, depicted in Fig. 3d, show that an $\alpha$ around 0.6 leads to good performance. Increasing $\alpha$ too much decreases the benefit of synthetic data generated from `DyDiff`.

## 6 CONCLUSION

In this paper, we explored the application of Diffusion Models (DMs) in sequence generation for decision-making problems, focusing on their role as dynamics models in fully offline reinforcement learning settings. We identified a critical issue where data directly synthesized by DMs can lead to a mismatch with the state-action distribution of the learning policy, negatively impacting policy learning. To address this, we introduced Dynamics Diffusion (`DyDiff`), a framework that effectively generates trajectories aligned with the learning policy's distribution, ensuring both policy consistency and dynamics accuracy of the synthetic trajectories. `DyDiff`'s superior performance stems from two critical components: (1) the intrinsic modeling ability of DMs and (2) the iterative correction mechanism between the DM and the learning policy. Both theoretical analysis and experiment results validate the effectiveness of these components. As an add-on scheme, `DyDiff` can be seamlessly integrated into any offline model-free algorithms that train explicit policies. Overall, `DyDiff` offers a promising direction for enhancing offline policy training using DMs. Furthermore, `DyDiff` holds potential for future extensions, including applications to online RL algorithms with more compact DM architectures since the training is relatively time-consuming with the full U-Net backbone, as well as approaches to improve scalability for large-scale tasks, which we aim to explore in future work.

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

## A  DETAILS OF THE MOTIVATION EXAMPLE

In this part, we list the details of experiment settings of our motivation example illustrated in Fig. 1.

For the first part (Fig. 1a), we randomly 5%/95% split the hopper-medium-replay dataset (Fu et al., 2020) into two parts, denoted as $\mathcal{D}_5$ adn $\mathcal{D}_{95}$, respectively. Then, we train a TD3BC (Fujimoto & Gu, 2021) agent on $\mathcal{D}_5$ while augmenting (1) on-policy data collected in the real environment; (2) data following the behavior policy randomly selected from $\mathcal{D}_{95}$; (3) no extra data to $\mathcal{D}_5$ every 50 epochs. We keep the data amount of scheme (1) and (2) the same for fair comparison. Note that both extra data in scheme (1) and (2) are real data without any error, and the only difference is that the former follows the distribution induced by the learning policy, whereas the latter follows the distribution induced by the behavior policy.

In the experiment about the rollout length (Fig. 1b), we also train the TD3BC agents on $\mathcal{D}_5$ and add model approximated on-policy data to it. For every epoch, we sample a batch of states from the dataset and start rollout from them. Though the rollout lengths differ, their transition amounts are kept the same by adjusting the state batch size. As single-step models cannot handle long-horizon rollout, we use `DyDiff` to do rollout in this experiment.

Finally, in Fig. 1c, we still train the TD3BC agents on $\mathcal{D}_5$ and add model approximated on-policy rollout trajectories of length 100. Those trajectories are synthesized by Bayesian Neural Networks (BNNs) suggested in MBPO (Janner et al., 2019) and `DyDiff`, respectively. For BNN, the trajectory is generated autoregressively as Eq. (6).

## B  DETAILS OF PRELIMINARIES

Both diffusion models and reinforcement learning contain the concept of step, which refers to the diffusion step in DMs and the timestep of trajectories in RL. To avoid confusion between them, we use the superscript to represent the diffusion step, whereas the subscript is for the RL timestep. For example, $x^i$ is the sample at the $i$-th diffusion step, and $s_t$ is the state at the $t$-th timestep in a RL trajectory.

### B.1  DIFFUSION MODEL

Diffusion models (DMs) are a class of generative models that mimic the diffusion process in physics. They first learn the data distribution and generate new data by incrementally removing noise from a pure Gaussian distribution. Formally, suppose the real data distribution is $p_{\text{data}}(x)$ and the initial sample is $x^0 \sim \mathcal{N}(0, I)$. For each timestep, DMs sample $x^{i+1} \sim p(x|x^{0:i})$. After $N$ timesteps, we obtain the final sample $x^N$, which is supposed to be distributed as $p_{\text{data}}(x)$. Therefore, the key point of DMs is to model and learn the distribution $p(x|x^{0:i})$. A widely used framework of DMs is DDPM (Ho et al., 2020), which formulates it as a parameterized Markov chain:

$$p_\theta(x^{0:N}) = p(x^0) \prod_{i=1}^{N} p_\theta(x^i|x^{i-1}), \quad p_\theta(x^i|x^{i-1}) = \mathcal{N}(\mu_\theta(x^{i-1}, i-1), \Sigma_\theta(x^{i-1}, i-1)) \quad (18)$$

The corresponding posterior $q(x^{0:N-1}|x^N)$ gradually adds Gaussian noise to the real data in a fixed variance schedule $\beta^i$:

$$q(x^{0:N-1}|x^N) = \prod_{i=1}^{N} q(x^{i-1}|x^i), \quad q(x^{i-1}|x^i) = \mathcal{N}(\sqrt{1-\beta^{i-1}}x^i, \beta^{i-1}I), \quad (19)$$

where $\beta^i$ is the hyperparameter. With the posterior distribution, DDPM learns $p_\theta$ by optimizing the variational lower bound:

$$\mathbb{E}[-\log p_\theta(x^N)] \leq \mathbb{E}_q \left[ -\log \frac{p_\theta(x^{0:N})}{q(x_{0:N-1}|x^N)} \right]. \quad (20)$$

After DDPM, many works propose variety of DDPM or improve the sample efficiency of DDPM (Song et al., 2020a; 2023; Nichol & Dhariwal, 2021). In this paper, we follow the architecture

proposed by EDM (Karras et al., 2022). EDM expresses DMs in a common framework by defining $p(x; \sigma)$ as the distribution obtained by adding Gaussian noise $\mathcal{N}(0, \sigma^2 I)$ to $p_{\text{data}}$. Let $\sigma_{\text{data}}$ be the standard deviation of $p_{\text{data}}$. If $\sigma_{\text{max}} \gg \sigma_{\text{data}}$, $p(x; \sigma_{\text{max}})$ becomes nearly the same as the pure Gaussian noise. Reversely, starting from a noise sample $x^0 \sim \mathcal{N}(0, \sigma_{\text{max}}^2 I)$, DMs denoise it following noise levels $\sigma_{\text{max}} = \sigma^0 > \sigma^1 > \cdots > \sigma^N = 0$. Finally, we obtain $x_N \sim p(x; \sigma^N) = p_{\text{data}}(x)$.

Following Song et al. (2020b), there is a corresponding probability flow ordinary differential equation (ODE) whose solution is our desired $p(x; \sigma)$:

$$\mathrm{d}x = -\dot{\sigma}(t)\sigma(t)\nabla_x \log p(x; \sigma(t))\mathrm{d}t \ . \tag{21}$$

Here, the noise level $\sigma(t)$ changes continuously with respect to time, $\dot{\sigma}(t) := \mathrm{d}\sigma(t)/\mathrm{d}t$, and $\nabla_x \log p(x; \sigma(t))$ is called the score function. As $t$ decreases, $x$ described by Eq. (21) will move towards the data distribution $p_{\text{data}}(x)$. Noting that $\sigma(t)$ is defined by ourselves, if the score function $\nabla_x \log p(x; \sigma(t))$ is known, we can sample $x$ by solving Eq. (21). Suppose $D_\theta(x; \sigma)$ is a denoiser function that predicts the real data from the noised sample $x$ and the noise level $\sigma$. Theoretical analysis shows that if $D_\theta$ minimizes the $L_2$ distance to $p_{\text{data}}$

$$\theta = \arg\min_\theta \mathbb{E}_{x \sim p_{\text{data}}} \mathbb{E}_{n \sim \mathcal{N}(0, \sigma^2 I)} \|D_\theta(x + n; \sigma) - x\|_2^2 \ , \tag{22}$$

then the score function can be expressed as

$$\nabla_x \log p(x; \sigma(t)) = \frac{D_\theta(x; \sigma) - x}{\sigma^2} \ . \tag{23}$$

For more detailed theoretical analysis and how to choose the noise level function $\sigma(t)$, please refer to the original paper of EDM (Karras et al., 2022).

## B.2 OFFLINE RL

Reinforcement learning (RL) models the sequential decision problem as a Markov Decision Process (MDP) $\mathcal{M} = (\mathcal{S}, \mathcal{A}, T, r, \gamma, d_0)$, where $\mathcal{S}$ is the state space and $\mathcal{A}$ is the action space. Let $\Delta(C)$ be the set of probability distributions over the set $C$. $T(s'|s, a): \mathcal{S} \times \mathcal{A} \to \Delta(\mathcal{S})$ is the dynamics function that gives the distribution over next state $s'$ when executing action $a$ at state $s$, $r(s, a): \mathcal{S} \times \mathcal{A} \to \mathbb{R}$ is the reward function, $\gamma \in (0, 1)$ is the discounted factor, and $d_0(s)$ is the distribution of the initial state. An agent on the MDP is a policy $\pi(a|s): \mathcal{S} \to \Delta(\mathcal{A})$ that defines a distribution over action $a$ given state $s$. The objective of RL is to learn a policy $\pi$ to maximize the discounted cumulative reward, as

$$\max_\pi J(\mathcal{M}, \pi) = \mathbb{E}_{s_0 \sim d_0, a_t \sim \pi(\cdot|s_t), s_{t+1} \sim T(\cdot|s_t, a_t)} \left[ \sum_{t=0}^\infty \gamma^t r(s_t, a_t) \right] \ . \tag{24}$$

In the online RL setting, the policy is allowed to interact with the environment, receiving real next states and rewards as feedback. However, such interaction is impractical in many real-world situations since it may be dangerous or cost a lot of resources. To address this problem, offline RL manages to train the policy $\pi$ on a pre-collected fixed dataset $\mathcal{D}_{\text{real}}$. The training objective of offline RL is the same as online RL given by Eq. (24), but the agent cannot receive real feedback to correct potential errors in training, which makes offline RL more challenging than online RL.

## C ALGORITHMS

We provide the overall algorithm of `DyDiff` in Algo. 1. To unify the notation in the initial rollout and the iteration, we define $\hat{\tau}_{a,\text{DM}}^{(0)} := \hat{\tau}_{a,\text{dyn}}$. Any diffusion sampling process that supports conditions can be incorporated for sampling the state sequence from $p_\theta$, and we choose the EDM sampler (Karras et al., 2022) for its high speed and accuracy.

For the sampling process, we slightly modify the EDM (Karras et al., 2022) sampling process to inject the first state $s_0$ and the action sequence $\tau_a$ as conditions.

The hyperparameters in Algo. 2 are the same as EDM. For those that should be adapted across datasets, we follow the grid search suggestion in Appendix E.2 of EDM (Karras et al., 2022) to find the best hyperparameters that minimize the loss of DMs. We list them and other hyperparameters used in training the DM in Tab. 3.

**Algorithm 1** DyDiff

---

**Require:** Offline dataset $\mathcal{D}$, number of training epochs $E$, number of optimization step $M$, rollout batch size $B_r$, ratio of real data $\alpha$, batch size $B$.

Train the DM $D_\theta(\tau; \sigma)$, the dynamics model $T_\phi(s, a)$, and the reward model $r_\psi(s, a)$ by Eq. (4), Eq. (5), Eq. (10), respectively.

Initial the synthetic replay buffer $\mathcal{D}_{\text{syn}} = \varnothing$ and the learning policy $\pi_\xi$.

**for** $e = 1 \rightarrow E$ **do**

    Sample a batch of state $\mathcal{B}_s = \{s_0^k\}_{k=1}^{B_r} \sim \mathcal{D}$ as initial states for rollout.

    **for** $s_0 \in \mathcal{B}_s$ **do**

        Autoregressively generate $\hat{\tau}_{\text{dyn}} = (s_0, \hat{a}_0, \hat{s}_1, \ldots, \hat{a}_{L-1}, \hat{s}_L)$ by $T_\phi$ and $\pi_\xi$.

        **for** $k = 1 \rightarrow M$ **do**

            Sample new trajectory $(s_0, \hat{\tau}_{s,\text{DM}}^{(k)}, \hat{\tau}_{a,\text{DM}}^{(k-1)}) \sim p_\theta(\tau|s_0, \hat{\tau}_{a,\text{DM}}^{(k-1)})$, following Algo. 2.

            Sample new action sequence $\hat{\tau}_{a,\text{DM}}^{(k)}$ from the learning policy $\pi_\xi$ by Eq. (8).

        **end for**

        Get final rollout trajectory $\hat{\tau}_{\text{DM}} := \hat{\tau}_{\text{DM}}^{(M)}$.

    **end for**

    Calculate the cumulative rewards $\{r_\psi(\hat{\tau}_{\text{DM}}^i)\}_{i=1}^{B_r}$.

    Filter the trajectories by their rewards using the hardmax or softmax filter.

    Add all transitions of remaining trajectories to $\mathcal{D}_{\text{syn}}$.

    Sample a batch of transitions $\mathcal{B}_{\text{syn}}$ from $\mathcal{D}_{\text{syn}}$, where $|\mathcal{B}_{\text{syn}}| = \lfloor \alpha B \rfloor$.

    Sample a batch of transitions $\mathcal{B}_{\text{real}}$ from $\mathcal{D}$, where $|\mathcal{B}_{\text{real}}| = B - |\mathcal{B}_{\text{syn}}|$.

    Use $\mathcal{B} = \mathcal{B}_{\text{real}} \cup \mathcal{B}_{\text{syn}}$ to train the learning policy $\pi_\xi$.

**end for**

**return** $\pi_\xi$

---

**Algorithm 2** Sampling process from the diffusion model

---

**Require:** Diffusion model $D_\theta(\tau; \sigma)$, diffusion step $N$, the first state $s_0$, action sequence $\tau_a$, timesteps $t_0, t_1, \ldots, t_N$, noise factors $\gamma_1, \gamma_2, \ldots, \gamma_{N-1}$, noise level $S_{\text{noise}}$.

Sample $\tau^0 \sim \mathcal{N}(0, t_0^2 \boldsymbol{I})$.

**for** $i = 0 \rightarrow N - 1$ **do**

    Sample $\epsilon_i \sim \mathcal{N}(0, S_{\text{noise}}^2 \boldsymbol{I})$.

    Increase the noise level $\hat{t}_i \leftarrow t_i + \gamma_i t_i$.

    Calculate $\hat{\tau}^i \leftarrow \tau^i + \sqrt{\hat{t}_i^2 - t_i^2} \epsilon_i$.

    Predict the denoised trajectories $\hat{\tau}^N = (\hat{s}_0^N, \hat{\tau}_{s>0}^N, \tau_a^N) \leftarrow D_\theta(\hat{\tau}^i; \hat{t}_i))$

    Evaluate the first-order gradient $\boldsymbol{d}_i \leftarrow (\hat{\tau}^i - \hat{\tau}^N)/\hat{t}_i$.

    Take the Euler step $\tau^{i+1} \leftarrow \hat{\tau}^i + (t_{i+1} - t_i)\boldsymbol{d}_i$.

    Apply hard replace $\tau^{i+1} \leftarrow (s_0, \tau_{s>0}^{i+1}, \tau_a)$.

    **if** $t_{i+1} \neq 0$ **then**

        $\boldsymbol{d}_i' \leftarrow (\tau^{i+1} - D_\theta(\tau^{i+1}; t_{i+1}))/t_{i+1}$.

        Apply the second order correction $\tau^{i+1} \leftarrow \hat{\tau}^i + (t_{i+1} - \hat{t}_i)(\boldsymbol{d}_i + \boldsymbol{d}_i')/2$.

        Apply hard replace $\tau^{i+1} \leftarrow (s_0, \tau_{s>0}^{i+1}, \tau_a)$.

    **end if**

**end for**

**return** $\tau^N$

---

Table 3: Hyperparameters used for training and sampling process following EDM.

| Hyperparameters | Values |
|---|---|
| $t_{i<N}$ | $\left(\sigma_{\max}^{1/\rho} + \frac{i}{N-1}(\sigma_{\min}^{1/\rho} - \sigma_{\max}^{1/\rho})\right)^{\rho}$ |
| $t_N$ | $0$ |
| $\gamma_{i<N}$ | $\begin{cases} \min\left(S_{\text{churn}}/N, \sqrt{2}-1\right) & \text{if } t_i \in [S_{\text{tmin}}, S_{\text{tmax}}] \\ 0 & \text{otherwise} \end{cases}$ |
| $\lambda(\sigma)$ | $(\sigma^2 + \sigma_{\text{data}}^2)/(\sigma * \sigma_{\text{data}})^2$ |
| $p_\sigma$ | $\ln \sigma \sim \mathcal{N}(P_{\text{mean}}, P_{\text{std}}^2)$ |
| $\sigma_{\min}$ | $0.002$ |
| $\sigma_{\max}$ | $80$ |
| $\sigma_{\text{data}}$ | $0.5$ |
| $\rho$ | $7$ |
| $S_{\text{tmin}}$ | $0.370$ |
| $S_{\text{tmax}}$ | $52.212$ |
| $S_{\text{churn}}$ | $60$ |
| $S_{\text{noise}}$ | $1.002$ |
| $P_{\text{mean}}$ | $-1.2$ |
| $P_{\text{std}}$ | $1.2$ |
| $N$ | $34$ |

# D    PROOFS

In this section, we provide proofs of lemmas and theories in the main paper.

## D.1    PROOF OF LEMMA 1

As Lemma 1 is from MBPO (Janner et al., 2019), we directly borrow the proof from MBPO with a slight modification. The following lemma from MBPO is necessary for proof.

**Lemma 2.** *(Lemma B.2 of MBPO). Suppose the error of a single-step dynamics model $T_m(s'|s,a)$ can be bounded as $\max_t \mathbb{E}_{a \sim \pi}[D_{\text{KL}}(T_m(s'|s,a)\|T(s'|s,a))] \leq \epsilon_m$. Then after executing the same policy $\pi$ from the same initial state $s_0$ for $t$ timesteps, the distance of the state marginal distribution at $s_t$ is bounded as*

$$D_{\text{TV}}(T_m(s_t|s_0,\pi)\|T(s_t|s_0,\pi)) \leq t\epsilon_m. \tag{25}$$

*Proof.* Let $\epsilon_t = D_{\text{TV}}(T_m(s_t|s_0,\pi)\|T(s_t|s_0,\pi))$. For brevity, we define $T_m^t(s) := T_m(s_t|s_0,\pi)$ and $T^t(s) := T(s_t|s_0,\pi)$.

$$
\begin{aligned}
|T_m^t(s) - T^t(s)| &= |\sum_{s'} T_m(s|s',\pi(s'))T_m^{t-1}(s') - T(s|s',\pi(s'))T^{t-1}(s')| \\
&\leq \sum_{s'} |T_m(s|s',\pi(s'))T_m^{t-1}(s') - T(s|s',\pi(s'))T^{t-1}(s')| \\
&\leq \sum_{s'} T_m^{t-1}(s')|T_m(s|s',\pi(s')) - T(s|s',\pi(s'))| + \sum_{s'} T(s|s',\pi(s'))|T_m^{t-1}(s') - T^{t-1}(s')| \\
&= \mathbb{E}_{s' \sim T_m^{t-1}(s')}[|T_m(s|s',\pi(s')) - T(s|s',\pi(s'))|] + \sum_{s'} T(s|s',\pi(s'))|T_m^{t-1}(s') - T^{t-1}(s')|
\end{aligned} \tag{26}
$$

$$
\begin{aligned}
\epsilon_t &= D_{\text{TV}}(T_m^t(s)\|T^t(s)) = \frac{1}{2}\sum_s |T_m^t(s) - T^t(s)| \\
&= \frac{1}{2}\sum_s \left( \mathbb{E}_{s' \sim T_m^{t-1}(s')}[|T_m(s|s',\pi(s')) - T(s|s',\pi(s'))|] + \sum_{s'} T(s|s',\pi(s'))|T_m^{t-1}(s') - T^{t-1}(s')| \right) \\
&= \frac{1}{2}\mathbb{E}_{s' \sim T_m^{t-1}(s')}\left[ \sum_s |T_m(s|s',\pi(s')) - T(s|s',\pi(s'))| \right] + D_{\text{TV}}(T_m^{t-1}(s')\|T^{t-1}(s')) \\
&\leq \epsilon_m + \epsilon_{t-1} \\
&= t\epsilon_m
\end{aligned} \tag{27}
$$

Then we can prove Lemma 1 following the original proof in MBPO.

**Lemma 1.** *(Lemma B.3 of MBPO). Suppose the error of a single-step dynamics model $T_m(s'|s,a)$ can be bounded as $\max_t \mathbb{E}_{a \sim \pi}[D_{\mathrm{KL}}(T_m(s'|s,a)\|T(s'|s,a))] \leq \epsilon_m$. Then after executing the same policy $\pi$ from the same initial state $s_0$ in $T_m$ and the real dynamics $T$, the expected returns are bounded as*

$$|J(T,\pi) - J(T_m,\pi)| \leq \frac{2R\gamma\epsilon_m}{(1-\gamma)^2} \ . \tag{28}$$

*Proof.* Denote the state-action distribution at timestep $t$ induced by $T$ as $p^t(s,a)$, and that by $T_m$ as $p_m^t(s,a)$.

$$
\begin{aligned}
|J(T,\pi) - J(T_m,\pi)| &= |\sum_{s,a}(p(s,a) - p_m(s,a))r(s,a)| \\
&\leq R|\sum_{s,a}\sum_t \gamma^t(p^t(s,a) - p_m^t(s,a))| \\
&\leq R\sum_t \gamma^t \sum_{s,a}|p^t(s,a) - p_m^t(s,a)| \\
&= 2R\sum_t \gamma^t D_{\mathrm{TV}}(p^t(s,a)\|p_m^t(s,a))
\end{aligned}
\tag{29}
$$

Note that $p^t(s,a) = T^t(s)\pi(a_t|s_t)$, which gives

$$D_{\mathrm{TV}}(p^t(s,a)\|p_m^t(s,a)) = D_{\mathrm{TV}}(T^t(s)\pi(a_t|s_t)\|T_m^t(s)\pi(a_t|s_t)) \leq D_{\mathrm{TV}}(T^t(s)\|T_m^t(s)) \ . \tag{30}$$

Therefore,

$$
\begin{aligned}
|J(T,\pi) - J(T_m,\pi)| &\leq 2R\sum_t \gamma^t D_{\mathrm{TV}}(T^t(s)\|T_m^t(s)) \\
&\leq 2R\sum_t \gamma^t t\epsilon_m \\
&= \frac{2R\gamma\epsilon_m}{(1-\gamma)^2}
\end{aligned}
\tag{31}
$$

### D.2    PROOF OF THEOREM 1

As Theorem 1 is similar with Lemma 1 with a slight modification in the assumption, we can prove Theorem 1 following the previous proof.

**Theorem 1.** *Suppose the error of a non-autoregressive model $T_d(s_t|s_0,\tau_a)$ can be bounded as $\max_t D_{\mathrm{TV}}(T_d(s_t|s_0,\tau_a))\|T(s_t|s_0,\tau_a) \leq \epsilon_d$. Then after executing the same policy $\pi$ from the same initial state $s_0$ in $T_d$ and the real dynamics $T$, the expected returns are bounded as*

$$|J(T,\pi) - J(T_d,\pi)| \leq \frac{2R\epsilon_d}{1-\gamma} \ . \tag{32}$$

*Proof.* The first part is the same as Eq. (29).

$$|J(T,\pi) - J(T_d,\pi)| \leq 2R\sum_t \gamma^t D_{\mathrm{TV}}(p^t(s,a)\|p_d^t(s,a)) \ . \tag{33}$$

Then, the non-autoregressive model gives a different state-action distribution as $p_d^t(s,a) = T_d(s_t|s_0,\tau_a)\pi(a_t|s_t)$, and the real distribution can be expressed as

$$
\begin{aligned}
p^t(s,a) &= T^t(s|s_0)\pi(a_t|s_t) \\
&= T^{t-1}(s'|s_0)T(s_t|s',a')\pi(a'|s')\pi(a_t|s_t) \\
&= \cdots \\
&= \pi(a_t|s_t)\prod_{j=1}^t T(s_j|s_{j-1},a_{j-1})\pi(a_{j-1}|s_{j-1}) \\
&= \pi(a_t|s_t)T(s_t|s_0,\tau_a)
\end{aligned}
\tag{34}
$$

Therefore, their TV distance is bounded by

$$D_{\text{TV}}(p^t(s,a)\|p_d^t(s,a)) \leq D_{\text{TV}}(T_d(s_t|s_0,\tau_a)\|T(s_t|s_0,\tau_a)) \,. \tag{35}$$

Following this, we can continue from Eq. (33):

$$
\begin{aligned}
|J(T,\pi) - J(T_d,\pi)| &\leq 2R \sum_t \gamma^t D_{\text{TV}}(p^t(s,a)\|p_d^t(s,a)) \\
&\leq 2R \sum_t \gamma^t D_{\text{TV}}(T_d(s_t|s_0,\tau_a)\|T(s_t|s_0,\tau_a)) \\
&\leq 2R \sum_t \gamma^t \epsilon_d \\
&= \frac{2R\epsilon_d}{1-\gamma}
\end{aligned}
\tag{36}
$$

### D.3    EMPIRICAL VALUES OF ERROR RATES

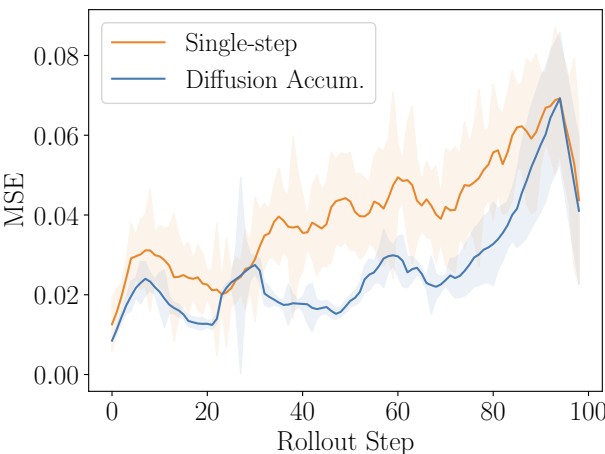

Figure 4: The transition-level MSE of single-step models and accumulative MSE of DMs for rollout, corresponding to $\epsilon_m$ and $\epsilon_d$, respectively.

To empirically validate our assumption that $\epsilon_m \approx \epsilon_d$, we conduct a rollout experiment using the `hopper-medium-replay` dataset with the TD3BC policy. We employ a pre-trained single-step dynamics model $T_m$ and a diffusion model $T_d$, alongside an expert TD3BC policy $\pi$. For each initial state $s_0$ sampled from the dataset, we first generate a rollout by having $\pi$ interact with $T_m$ autoregressively, following the scheme described in the main paper. Let $\tau_m = (\tau_{s,m}, \tau_a)$ denote this trajectory. Next, $s_0$ and $\tau_a$ are fed in to the DM $T_d$ to synthesize a new rollout $\tau_d = (\tau_{s,d}, \tau_a)$. Finally, we execute $\tau_a$ from $s_0$ in the real environment, obtaining the ground truth trajectory $\tau = (\tau_s, \tau_a)$. As the action is consistent across all three rollouts, we focus on computing the MSE of the state sequence, as:

$$e_{m,t} = \|s_{m,t} - s_t\|_2^2, \quad e_{d,t} = \|s_{d,t} - s_t\|_2^2 \,. \tag{37}$$

The estimated transition-level MSE $e_{m,t}$ reflects the error rate of the single-step dynamics model $\epsilon_m$. In contrast, the error rate of the DM is defined by executing a $t$-step action sequence, estimated by $E_{d,t} = \sum_{i=1}^t e_{d,i}$.

We repeat the experiment over multiple initial states and random seeds, plotting $e_{m,t}$ and $E_{d,t}$ over $t$, as shown in Fig. 4. The results demonstrate that $E_{d,t} < e_{m,t}$ over a long horizon, supporting our assumption that $\epsilon_d \approx \epsilon_m$. Notably, comparing the accumulative error $E_{d,t}$ against the single-step error $e_{m,t}$ further demonstrates the superior long-horizon generation capability of DMs.

### D.4    EXPLANATION TO ASSUMPTIONS

To illustrate the effectiveness of the iteration process in `DyDiff`, we first introduce Assumption 1 and Assumption 2. Here, we provide an intuitive explanation for these two assumptions.

The Assumption 1 can be decomposed into two assumptions:

**Assumption 3.** *The error between $T(s_t|s_0, \tau_a)$ and $T_d(s_t|s_0, \tau_a)$ can be bounded as* $\max_t D_{\mathrm{TV}}(T_d(s_t|s_0, \tau_a)\|T(s_t|s_0, \tau_a)) \leq \epsilon_{s,d}$*, where $\epsilon_{s,d}$ is a constant.*

**Assumption 4.** *The error between $T_d(s_t|s_0, \tau_a)$ and $T_d(s_t|s_0, \tau_{a,d})$ can be bounded as* $\max_t D_{\mathrm{TV}}(T_d(s_t|s_0, \tau_{a,d})\|T_d(s_t|s_0, \tau_a)) \leq C_{a,d} \max_t \|\tau_{a,d} - \tau_a\|$*, where $C_{a,d}$ is a constant.*

With Assumption 3 and Assumption 4, the Assumption 1 is actually a corollary. Using the triangular inequality of the TV distance, we have

$$
\begin{aligned}
\max_t D_{\mathrm{TV}}(T_d(s_t|s_0, \tau_{a,d})\|T(s_t|s_0, \tau_a)) &\leq \max_t [D_{\mathrm{TV}}(T_d(s_t|s_0, \tau_{a,d})\|T_d(s_t|s_0, \tau_a)) \\
&\quad + D_{\mathrm{TV}}(T_d(s_t|s_0, \tau_a)\|T(s_t|s_0, \tau_a))] \\
&\leq \max_t D_{\mathrm{TV}}(T_d(s_t|s_0, \tau_{a,d})\|T_d(s_t|s_0, \tau_a)) \\
&\quad + \max_t D_{\mathrm{TV}}(T_d(s_t|s_0, \tau_a)\|T(s_t|s_0, \tau_a)) \\
&\leq C_{a,d} \max_t \|\tau_{a,d} - \tau_a\| + \epsilon_{s,d} \ .
\end{aligned}
\tag{38}
$$

The Assumption 3 is the same as the condition of Theorem 1. For Assumption 4 and Assumption 2, their forms are similar to the Lipschitz condition. Assumption 4 bounds the change in the state distribution induced by the diffusion model when the action sequence changes, whereas Assumption 2 bounds the change in the action distribution induced by the learning policy when the state changes. In practice, when input states and actions do not fall far from the data coverage of the training set, these assumptions can be assumed to hold. In the far-out-of-distribution region, the accuracy of models becomes too low for us to predict their behavior, where these assumptions are probably violated.

# E  EXPERIMENTS

In this section, we list the detailed settings of `DyDiff` for experiments, and comparison between hardmax and softmax filters.

## E.1  EXPERIMENT DETAILS

We implement `DyDiff` under the ILSwiss[1] framework, which provides RL training pipelines in PyTorch. As an add-on scheme over offline policy training algorithms, we reimplement the base algorithms over our codebase, and we refer to their official implementations from:

- TD3BC: `https://github.com/sfujim/TD3_BC`
- CQL: `https://github.com/aviralkumar2907/CQL`
- DiffQL: `https://github.com/Zhendong-Wang/Diffusion-Policies-for-Offline-RL`

The additional hyperparameters of `DyDiff` are listed in Tab. 4. We do not change the hyperparameters of the underlying policy training algorithms, thus they are omitted here.

## E.2  ABLATION STUDIES ON FILTER TYPE

We propose two filter schemes: the hardmax filter and the softmax filter in Section 4.2. For further comparison, we test both filters on MuJoCo locomotion tasks and over all base policies, and the results are listed in Tab. 5. It shows that `DyDiff`-H and `DyDiff`-S have no significant performance gap when the data coverage is relatively narrow such as `md` dataset, but the hardmax filter is slightly worse on `mr` and `me` datasets. A possible reason is that the softmax filter will provide more diversified data, which are easy to go outside of the data coverage, reducing the data accuracy. We suggest using the softmax filter as the default.

To examine whether the filtering scheme enhances the performance of SynthER, we apply the same softmax filter to the data generated by SynthER. Since SynthER synthesizes transitions rather than

___
[1]`https://github.com/Ericonaldo/ILSwiss`

Table 4: Additional hyperparameters for `DyDiff`.

| Hyperparameters | Values |
|---|---|
| Batch size $B$ | 256 |
| Rollout batch size $B_r$ | 2048 |
| Real ratio $\alpha$ | 0.6 |
| Rollout length $L$ | 100 |
| Iteration time $M$ | 2 (MuJoCo locomotion) 
 1 (Maze2D) |
| Filter proportion $\eta$ | 0.8 (`mr` and `me`) 
 0.6 (`md` and Maze2D) |
| Softmax temperature | 0.05 |

Table 5: Full results on MuJoCo locomotion tasks that include both hardmax and softmax filters. `DyDiff` with hardmax filter is denoted as `DyDiff`-H, whereas that with softmax filter as `DyDiff`-S.

| Dataset | TD3BC | | | | CQL | | | | DiffQL | | | |
|---|---|---|---|---|---|---|---|---|---|---|---|---|
| | Base | SynthER | DyDiff-H | DyDiff-S | Base | SynthER | DyDiff-H | DyDiff-S | Base | SynthER | DyDiff-H | DyDiff-S |
| hopper-md | 65.8±5.8 | 59.0±5.2 | 52.2±3.6 | 71.5±15.5 | 57.9±3.7 | 57.1±2.3 | 54.1±2.0 | 54.9±2.3 | 61.0±5.6 | 58.9±4.8 | 58.2±4.5 | 58.6±4.9 |
| hopper-me | 95.2±14.9 | 86.1±7.6 | 94.5±14.1 | 98.4±13.4 | 85.3±9.8 | 92.3±7.4 | 88.4±10.2 | 90.9±8.2 | 106.7±6.3 | 108.2±4.8 | 107.1±2.7 | 109.2±3.0 |
| hopper-mr | 81.5±17.4 | 46.3±7.7 | 93.5±22.7 | 82.6±20.1 | 87.7±7.8 | 92.4±6.5 | 87.8±8.0 | 95.3±2.6 | 97.8±5.1 | 99.1±4.4 | 99.5±2.0 | 99.5±3.4 |
| halfcheetah-md | 50.6±0.5 | 51.2±2.9 | 57.4±3.8 | 58.9±2.1 | 43.8±2.6 | 43.7±0.2 | 43.1±0.2 | 43.2±1.1 | 47.1±2.5 | 47.3±2.6 | 47.6±2.7 | 47.5±2.8 |
| halfcheetah-me | 69.7±18.4 | 87.0±8.1 | 87.0±8.1 | 77.6±10.6 | 53.0±9.0 | 49.4±5.1 | 65.0±13.2 | 60.8±9.2 | 94.2±3.0 | 90.2±4.7 | 93.0±4.2 | 92.6±5.7 |
| halfcheetah-mr | 46.0±0.6 | 46.7±2.7 | 45.6±6.0 | 44.2±6.1 | 42.9±2.6 | 43.2±0.3 | 41.5±0.3 | 41.5±2.2 | 39.5±8.5 | 46.0±2.8 | 47.1±2.9 | 46.6±2.5 |
| walker2d-md | 76.8±16.3 | 8.0±7.4 | 68.6±14.3 | 87.9±1.1 | 79.3±2.4 | 82.5±1.1 | 78.5±0.3 | 79.4±0.2 | 84.4±0.6 | 85.0±1.3 | 83.2±1.9 | 82.7±1.9 |
| walker2d-me | 110.7±0.6 | 111.7±0.6 | 107.0±6.8 | 110.6±1.3 | 108.9±0.6 | 109.1±0.4 | 107.8±0.2 | 108.8±0.4 | 109.6±0.2 | 109.8±0.4 | 109.9±0.2 | 109.9±0.4 |
| walker2d-mr | 85.8±11.8 | 91.9±6.1 | 28.4±21.5 | 74.5±8.9 | 80.5±3.7 | 85.7±2.8 | 84.5±4.9 | 86.8±7.0 | 90.6±1.9 | 94.4±3.5 | 92.1±2.6 | 92.3±2.2 |
| Average | 75.8 | 65.3 | 70.5 | **79.6** | 71.0 | 72.8 | 72.3 | **73.5** | 81.2 | **82.1** | 82.0 | **82.1** |

entire trajectories, the softmax filter is applied at the transition level. Specifically, we calculate the softmax rewards of synthetic transitions to determine their sampling probabilities and select the same proportion, $\eta$, of these transitions for training TD3BC agents. The results, presented in Tab. 6, indicate that the reward filter yields a slight improvement in performance compared to the original SynthER. However, the performance gains are primarily observed in relatively simple tasks, such as Hopper and Walker2d. Conversely, filtered SynthER underperforms relative to the original SynthER on more complex tasks like HalfCheetah and Maze2d-large. This may occur because selecting high-reward transitions limits the training data to better but less accessible regions, which does not necessarily benefit policy learning. For `DyDiff`, we apply the filtering scheme at the trajectory level, preserving the complete paths leading to high-reward regions.

### E.3 ANALYSIS OVER TASKS AND DATASET TYPES

To better understand the advantages and limitations of `DyDiff`, we compute the normalized interquartile mean (IQM) scores as suggested by Agarwal et al. (2021), grouped by environment and dataset type. For the IQM scores, we evaluate the trained policy in the real environment, exclude the top 25% and bottom 25% of results, and compute the mean of the remaining data. This statis-

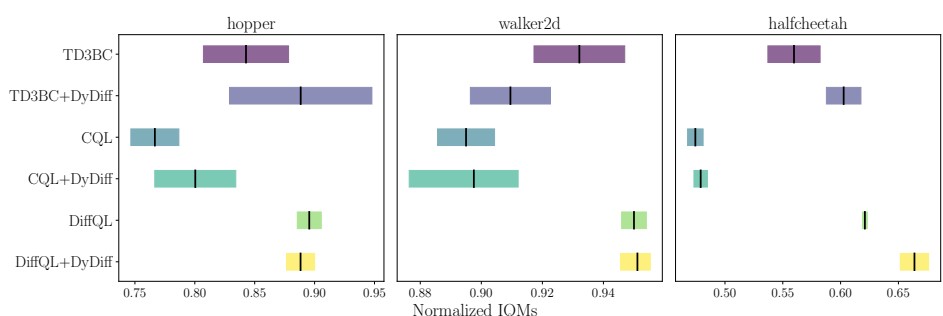

Figure 5: Normalized IQM scores grouped by the environment.

Table 6: Results in comparison to filtered SynthER (SynthER-f) on MuJoCo locomotion tasks and Maze2D navigation tasks, with the underlying policy TD3BC. The best average results are in **bold**.

| Dataset | TD3BC | | | |
| --- | --- | --- | --- | --- |
| | Base | SynthER | SynthER-f | **DyDiff** |
| hopper-md | 65.8±5.8 | 59.0±5.2 | 62.9±3.4 | 71.5±15.5 |
| hopper-me | 95.2±14.9 | 94.1±12.3 | 96.6±11.5 | 98.4±13.4 |
| hopper-mr | 81.5±17.4 | 50.4±13.4 | 51.4±19.8 | 82.6±20.1 |
| halfcheetah-md | 50.6±0.5 | 51.2±2.9 | 48.3±0.4 | 58.9±2.1 |
| halfcheetah-me | 69.7±18.4 | 80.0±7.5 | 78.7±8.0 | 77.6±10.6 |
| halfcheetah-mr | 46.0±0.6 | 45.2±0.4 | 43.6±0.3 | 44.2±6.1 |
| walker2d-md | 76.8±16.3 | 83.5±2.1 | 84.5±2.2 | 87.9±1.1 |
| walker2d-me | 110.7±0.6 | 110.6±0.4 | 110.5±0.6 | 110.6±1.3 |
| walker2d-mr | 85.8±11.8 | 90.4±5.3 | 91.1±3.0 | 74.5±8.9 |
| **Average** | 75.8 | 73.8 | 74.2 | **79.6** |
| maze2d-umaze | 0.35±0.10 | 0.32±0.09 | 0.39±0.15 | 0.55±0.12 |
| maze2d-medium | 0.81±0.50 | 0.49±0.20 | 0.73±0.28 | 1.34±0.19 |
| maze2d-large | 0.43±0.46 | 0.98±0.33 | 0.87±0.29 | 1.82±0.42 |
| Average | 0.53 | 0.60 | 0.66 | **1.24** |

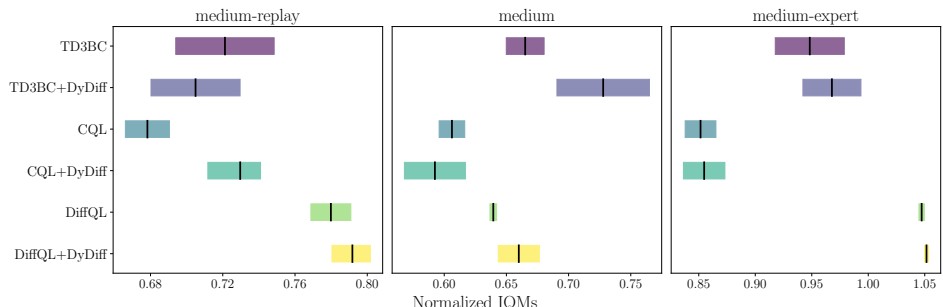

Figure 6: Normalized IQM scores grouped by the dataset type.

tical approach mitigates the impact of outliers on the final results. Using IQMs, we observe that DyDiff shows slight instability in walker2d, particularly in walker2d-mr. This instability likely stems from the walker2d-mr dataset containing a large amount of low-quality data, reducing the accuracy of rollouts generated by DyDiff. On the contrary, DyDiff performs well in medium-expert datasets, suggesting that the synthetic data are both accurate and of high rewards. Overall, incorporating DyDiff tends to improve the performance of underlying model-free policies.

### E.4 COMPARISON TO MTDIFF-S

MTDiff (He et al., 2024) utilizes DMs as the planner or the data synthesizer to solve offline multi-task RL problems. It proposes two variants of MTDiff: MTDiff-p directly plans the future trajectories and selects the action to be executed, while MTDiff-s only synthesizes extra data to assist policy training. We compare DyDiff with MTDiff-s on single-task datasets with the underlying policy TD3BC, and the results are listed in Tab. 7. Note that MTDiff-s is originally designed to solve multi-task problems, where the DM can learn knowledge across different tasks and generalize to unseen tasks. In single-task scenarios, MTDiff-s does not leverage its full potential, thus only reaching similar performance as SynthER, and is worse than DyDiff.

Table 7: Results in comparison to MTDiff-s on MuJoCo locomotion tasks and Maze2D navigation tasks, with the underlying policy TD3BC. The best average results are in **bold**.

| Dataset | TD3BC | | | |
|---|---|---|---|---|
| | Base | SynthER | MTDiff-s | **DyDiff** |
| `hopper-md` | 65.8±5.8 | 59.0±5.2 | 55.1±3.3 | 71.5±15.5 |
| `hopper-me` | 95.2±14.9 | 94.1±12.3 | 85.2±10.1 | 98.4±13.4 |
| `hopper-mr` | 81.5±17.4 | 50.4±13.4 | 78.4±12.4 | 82.6±20.1 |
| `halfcheetah-md` | 50.6±0.5 | 51.2±2.9 | 46.7±2.6 | 58.9±2.1 |
| `halfcheetah-me` | 69.7±18.4 | 80.0±7.5 | 71.2±8.3 | 77.6±10.6 |
| `halfcheetah-mr` | 46.0±0.6 | 45.2±0.4 | 43.3±0.5 | 44.2±6.1 |
| `walker2d-md` | 76.8±16.3 | 83.5±2.1 | 82.0±1.0 | 87.9±1.1 |
| `walker2d-me` | 110.7±0.6 | 110.6±0.4 | 110.4±0.5 | 110.6±1.3 |
| `walker2d-mr` | 85.8±11.8 | 90.4±5.3 | 80.4±4.8 | 74.5±8.9 |
| **Average** | 75.8 | 73.8 | 72.5 | **79.6** |
| `maze2d-umaze` | 0.35±0.10 | 0.32±0.09 | 0.31±0.06 | 0.55±0.12 |
| `maze2d-medium` | 0.81±0.50 | 0.49±0.20 | 0.61±0.20 | 1.34±0.19 |
| `maze2d-large` | 0.43±0.46 | 0.98±0.33 | 0.86±0.31 | 1.82±0.42 |
| Average | 0.53 | 0.60 | 0.59 | **1.24** |

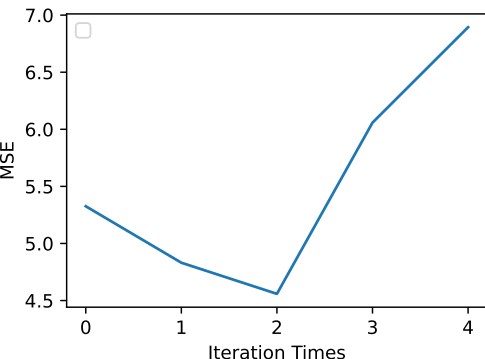

Figure 7: Change of the total MSE of synthetic trajectories over the iteration times.

### E.5 SYNTHETIC ERROR WITH ITERATION TIMES

In practice, the iteration times $M$ cannot be arbitrarily large since the intermediate result may go out of the data distribution of the dataset, which significantly increases the error of DM generation. As an illustrative example, we compute the total MSE of generated trajectories during the generation process and plot how it changes over the iteration times, shown in Fig. 7. We test it in the `hopper-medium-replay` task with a TD3BC policy, and the single-step dynamics model and the diffusion model are the same as we used in the main experiments. The results show that the initial MSE of trajectories generated by the single-step dynamics is relatively large. After two steps of refinement by the DM and the learning policy, the MSE decreases but rapidly goes up as the iteration continues. In practice, using $M = 1$ or 2 is sufficient for accurate generation.

### E.6 VISUALIZATION ON MAZE2D

To further investigate how the quality of the single-step dynamics model and the learning policy affect the synthetic trajectories in `DyDiff`, we visualize the trajectories in Maze2D-medium, as shown in Fig. 8. For each setting, we sample 64 initial states from the dataset and generate rollouts starting from them. Fig. 8(a)(b)(c) utilizes a random single-step dynamics, while (d)(e)(f) are with a trained single-step dynamics the same as the main paper. For quality of policies, (a)(d) tests random policies,

(b)(e) medium-level policies, and (c)(f) expert policies. In each subfigure, the left maze depicts the trajectories generated autoregressively by the policy and the single-step dynamics, and the right one shows those after one-step refinement by `DyDiff`. Generally, `DyDiff` can optimize the quality of trajectories with various dynamics models and policies. Comparing to the trained single-step dynamics, we find that the single-step dynamics is prone to omitting the obstacles in the maze, while most trajectories refined by `DyDiff` bypass the walls. Although the single-step dynamics can learn the real dynamics in this simple task, it fails to learn the general distribution. On the contrary, the modeling ability of DMs allows `DyDiff` to learn the knowledge of obstacles from the long-horizon data distribution.

We also illustrate how the synthetic trajectories change over the refinement iteration in Fig. 9, with a trained single-step dynamics and the medium-level policy. We annotate the number of legal trajectories after each iteration. Here, a trajectory is legal if it does not contain states in the wall. This results also support our observation that the single-step dynamics model cannot learn long-horizon distribution, providing more illegal trajectories, and the iterative refinement of DMs will improve the data quality.

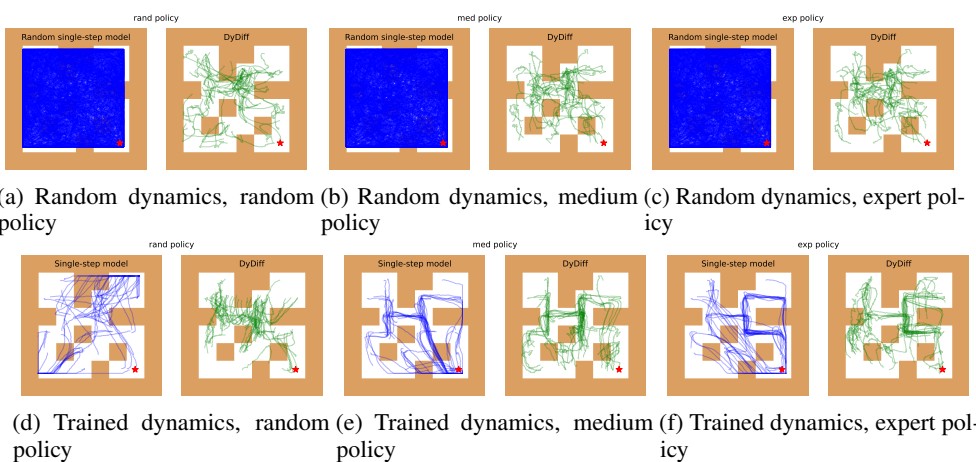

(a) Random dynamics, random policy    (b) Random dynamics, medium policy    (c) Random dynamics, expert policy

(d) Trained dynamics, random policy    (e) Trained dynamics, medium policy    (f) Trained dynamics, expert policy

Figure 8: Synthetic trajectories in Maze2D-medium from different single-step dynamics and policies.

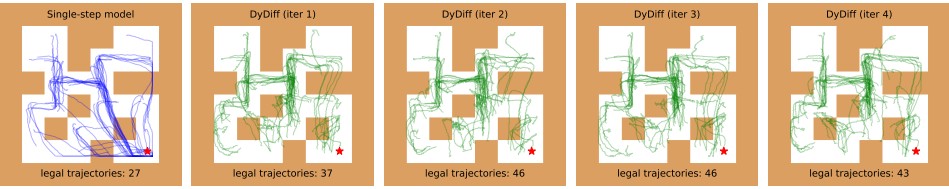

Figure 9: The change of synthetic trajectories over the refinement iteration in Maze2D.

### E.7 COMPUTATIONAL RESOURCES AND MODEL SIZES

Most experiments are conducted on NVIDIA RTX 3080 Ti GPUs. The training time of `DyDiff` is about 20 hours in addition to the original time cost of the underlying policies for each task. In comparison, training a SynthER model and generating $5 \times 10^6$ samples cost about 2.5 hours. Also, we would like to point out that the training time in offline RL is usually less important than that in online RL. For deployment, the DM is no longer used once the policy training is finished, so the inference time depends on the specific underlying RL algorithms themselves.

As for model sizes, `DyDiff` leverages the same DM structure as EDM, which is about 58M, whereas SynthER is 6.5M.

