# OpenReview forum: "DyDiff: Long-Horizon Rollout via Dynamics Diffusion for Offline Reinforcement Learning"
_ICLR.cc/2025/Conference — Submitted to ICLR 2025_

### Official Review · Reviewer_kq5G · 2024-10-18

**Soundness:** 3
**Presentation:** 3
**Contribution:** 2
**Rating:** 6
**Confidence:** 4

**Summary:**

The paper proposed Dynamics Diffusion (DyDiff) to improve offline RL by augmenting the dataset with a single-step dynamics model followed by an iterative refinement process.
DyDiff demonstrates better results on MuJoCo locomotion tasks with ablation studies.

**Strengths:**

- The refinement process combining policy and trajectory diffusion models is novel.
- Theoretical analysis is provided to support the method and results.
- The paper is well-written and easy to follow.

**Weaknesses:**

- The performance improvement is minor (within standard deviation) in most of the datasets.
- The method trains a single-step dynamics model to seed the trajectory diffusion model. The application is limited when the transition distribution coverage in the dataset isn't large enough to train the dynamics model.

**Questions:**

- How does the accuracy of the single-step dynamics model affect the final performance? How would sub-optimal initial policies and dynamics models impact the result?
Are there some visualizations of how the rollouts evolve across different iterations? I recommend showing some visualization results to show how the iterative process improves the rollouts. It is hard to tell if the improvement comes from the iteration or other components with current results.
- In Figure 3a, the method seems sensitive to the iteration time $M$. How could one determine a good value by observing the dataset and the environment?
- In section 5.3, why does DyDiff improve much more than locomotion tasks? Does the improvement come from sparse rewards or easier dynamics in the environments?
- What is the training resource of DyDiff compared with baselines in terms of time and model sizes?

Overall, I have concerns about DyDiff's ability to capture the dynamics when the environments and datasets varied and what is the respective contribution of the seeding trajectory and the iterative process. I'm open to raising my score if the concerns are addressed.

---

> ### Author Response · Authors · 2024-11-21
>
> - Q1: How does the accuracy of the single-step dynamics model affect the final performance? How would sub-optimal initial policies and dynamics models impact the result? Are there some visualizations of how the rollouts evolve across different iterations? I recommend showing some visualization results to show how the iterative process improves the rollouts. It is hard to tell if the improvement comes from the iteration or other components with current results.
> - A1: Thanks for your valuable question. We visualize the synthetic trajectories in Maze2D tasks with different accuracy of the single-step dynamics model and the learning policy, as well as how they evolve across different iterations. Please refer to Appendix E.6 for the visualization results and detailed analysis.
>
> - Q2: In Figure 3a, the method seems sensitive to the iteration time $M$. How could one determine a good value by observing the dataset and the environment?
> - A2: Emperically, the choice of $M$ depends on the complexity of the environment. For example, we choose $M=2$ for MuJoCo locomotion tasks, while $M=1$ for Maze2D tasks with relatively simple dynamics. In practice, we find that using larger $M$ will not necessarily decrease the performance but definitely cost more training time. So searching $M=1,2$ is sufficient for most cases.
>
> - Q3: In section 5.3, why does DyDiff improve much more than locomotion tasks? Does the improvement come from sparse rewards or easier dynamics in the environments?
> - A3: We would like to contribute this to the modeling ability of long-horizon distributions. As the visualization results in Appendix E.6 show, both simple single-step MLP models and DyDiff can capture the dynamics in Maze2D tasks. However, the single-step model can hardly realize the obstacles, providing many trajectories going through the walls. These trajectories are correct in dynamics if we omit the wall, but they obviously do harm to the policy learning. In contrast, DyDiff can model the long-horizon distribution using diffusion models and synthesize most legal trajectories. This comparison proves the improvement comes from not only the easier dynamics but also the modeling ability of DyDiff.
>
> - Q4: What is the training resource of DyDiff compared with baselines in terms of time and model sizes?
> - A4: Thanks for your advice. We have provided the training time and the model sizes of DyDiff and SynthER in Appendix E.7. As an add-on method, DyDiff does not affect the inference time of the underlying policies.

---

> > ### Comment · Reviewer_kq5G · 2024-11-25
> >
> > The revision addresses most of my concerns. However, DyDiff requires a lot more computing resources but only minor performance improvement. I will consider a weak accept.

---

### Official Review · Reviewer_3Lt8 · 2024-10-28

**Soundness:** 2
**Presentation:** 3
**Contribution:** 3
**Rating:** 6
**Confidence:** 4

**Summary:**

In this work, the authors propose a novel model-based extension applicable to many existing offline Reinforcement Learning algorithms. Specifically, the authors propose utilizing both a single-step dynamics model and a diffusion model for trajectory modelling to extend the offline dataset based on on-policy rollouts. In their method, the single-step dynamics model together with the learned policy is used to generate a trajectory rollout from a state contained in the dataset. The actions selected by the policy during this rollout are then used to condition the diffusion model which predicts corresponding trajectory states with a lower error than the single-step model. Using an iterative scheme between predicting state trajectories, using the diffusion model, and corresponding actions, using the policy, the method generates new state-action-trajectories to be added to the training dataset. A learned reward model is employed to augment the trajectory with rewards and filter out low reward trajectories. The paper contains both experimental as well as theoretical analysis of the proposed method.

**Strengths:**

•	The motivational example in Figure 1 is well done.

•	Nicely written Related Work, giving a good overview over both diffusion in RL and different model-based approaches in offline RL.

•	I like the more informal style of writing, even though clarity could be improved at points.

•	Nice overview in Figure 2. However, I’d suggest adding numbers 1), 2), 3) also to the figure to ease navigating the sketch. Also, variable k is not clear from the figure alone.

**Weaknesses:**

•	Preliminaries are sufficient for a reader familiar with the field but might be brief for readers unfamiliar with either Diffusion Models or RL.

•	The notation in Section 4, especially the trajectory superscript, confused me. It appears the authors use the superscript both for the diffusion step i and “trajectory improvement iteration” k. If I conclude correctly, the superscript is in parenthesis if it regards the trajectory improvement iteration. However, this is not consistent, e.g. from line 274 on in section 4.2. Perhaps a brief introduction to the notation, e.g. in section 4.1, and more consistency could help.

•	I would have liked to see a bit more on what this paper contributes in comparison to concurrent work of Jackson et al.

•	The results in Section 5.2 indicate that DyDiff improves the performance in some settings, decreases performance in other settings and seems almost identical in the rest of settings. Without any significance tests the claim that “DyDiff significantly improves the performance of these algorithms without any additional hyperparameter tuning.” seems over the top. Please provide significance tests and/or rephrase the claim to be more precise.

•	Similar to the results in 5.2, the ablation studies in 5.4 seem to have little reliability due to high variance.

**Questions:**

•	In line 232 you state that “Though we can now use DMs to generate state trajectories, the initial action trajectory for the condition is still left blank.” while Figure 2 already told the reader that the initial action sequence is initialized with the actions from the policy during one-step rollouts. For me the lines 182 to 186 thus caused a lot of confusion, which was then resolved later when reading section 4.2. Is there a particular reason for this structure or could it be adapted?

•	Rather a suggestion: The caption of Figure 2, and the corresponding description in lines 181-186 could be more clear.

•	In Section 5.2 you state that “From the perspective of different base policies, DyDiff exhibits relative incompatibility with CQL”, which seems intuitive, but nonetheless, CQL seems to have similar benefit from DyDiff than the other baselines in Section 5.3. Do you have an intuition on this?

--

Post-rebuttal: While I believe the current revision resolves most of the clarity issues, the reported probabilities of improvement, ranging from 0.52 to 0.6, are not convincing enough to warrant a more favorable evaluation as a "good paper." Therefore, I will maintain my initial evaluation of 6 (weak accept).

---

> ### Author Response · Authors · 2024-11-21
>
> - Q1: Preliminaries are sufficient for a reader familiar with the field but might be brief for readers unfamiliar with either Diffusion Models or RL.
> - A1: Thanks for your advice. We have added a section to provide detailed preliminaries in Appendix B.
>
> - Q2: The notation in Section 4, especially the trajectory superscript, confused me. It appears the authors use the superscript both for the diffusion step i and “trajectory improvement iteration” k. If I conclude correctly, the superscript is in parenthesis if it regards the trajectory improvement iteration. However, this is not consistent, e.g. from line 274 on in section 4.2. Perhaps a brief introduction to the notation, e.g. in section 4.1, and more consistency could help.
> - A2: We apologize for possible confusion caused by our notations. There are two types of timesteps in Diffusion Models for RL, i.e., the timestep of states and actions in RL trajectories and the timestep in the diffusion process. To represent the iteration times of DyDiff, we have to use parenthesis superscripts. We add a brief introduction in Section 4.2, and fixed confused notations from Line 274, marked in red.
>
> - Q3: I would have liked to see a bit more on what this paper contributes in comparison to concurrent work of Jackson et al.
> - A3: The work PGD[1] identifies a similar policy mismatch problem in applying diffusion synthetic data to RL policy training. It computes the log-likelihood of synthetic trajectories related to the learning policy, which is then used as guidance in the denoising process. In contrast, DyDiff addresses this problem using a different approach without guidance, thus can work even though the log-likelihood is intractable. For example, TD3BC trains a deterministic policy, so PGD has to re-model it as a unit Gaussian distribution to obtain the log-likelihood, while DyDiff can be applied to it seaminglessly. Besides, we develop theoretical analysis about the generation ability of general DMs.
>
> - Q4: The results in Section 5.2 indicate that DyDiff improves the performance in some settings, decreases performance in other settings and seems almost identical in the rest of settings. Without any significance tests the claim that “DyDiff significantly improves the performance of these algorithms without any additional hyperparameter tuning.” seems over the top. Please provide significance tests and/or rephrase the claim to be more precise. Similar to the results in 5.2, the ablation studies in 5.4 seem to have little reliability due to high variance.
> - A4: Thanks for your advice, and we have removed the word "significantly". Also, we compute the probability of improvement (PI) following RLiable[2] for our main experiments. Note that DyDiff is an additional method to underlying model-free policies, so we only compare DyDiff+X with its counterpart X. The results are listed below.
>
> | Algorithm | P(DyDiff+TD3BC>TD3BC) | P(DyDiff+CQL>CQL) | P(DyDiff+DiffQL>DiffQL) |
> | -------- | -------- | -------- | -------- |
> | PI     | 0.60 (0.41-0.78) | 0.52 (0.37-0.68)  |0.59 (0.32-0.85) |
>
>   We would like to point out that in offline RL, the trained policies are usually more unstable than those in online RL because they have no chance to receive real feedback to correct themselves. Therefore, the variance of evaluation results is usually high. However, with proper off-policy evaluation strategies, one can select the good policies for real deployment.
>
> - Q5: In line 232 you state that “Though we can now use DMs to generate state trajectories, the initial action trajectory for the condition is still left blank.” while Figure 2 already told the reader that the initial action sequence is initialized with the actions from the policy during one-step rollouts. For me the lines 182 to 186 thus caused a lot of confusion, which was then resolved later when reading section 4.2. Is there a particular reason for this structure or could it be adapted? Rather a suggestion: The caption of Figure 2, and the corresponding description in lines 181-186 could be more clear.
> - A5: We apologize for confusion caused by our ambiguous description in Figure 2, Line 181-186, and the last paragraph of Section 4.1. In our original writing, although we have introduced the main structure of the algorithm, we still write as if the reader did not know how the initial action sequence is obtained. We have revised the text in the caption of Figure 2, Line 181-186, and the last paragraph of Section 4.1, marked in red.

---

> ### Author Response · Authors · 2024-11-21
> **Rebuttal Cont.**
>
> - Q6: In Section 5.2 you state that “From the perspective of different base policies, DyDiff exhibits relative incompatibility with CQL”, which seems intuitive, but nonetheless, CQL seems to have similar benefit from DyDiff than the other baselines in Section 5.3. Do you have an intuition on this?
> - A6: We would like to contribute this improvement to the modeling ability on long-horizon distributions, which is much more important in sparse-reward tasks than dense-reward ones. In Maze2D, the agent is not rewarded until it reaches the goal. DyDiff can synthesize high-quality long-horizon rollouts that can help the policy find the path to the goal quickly. Therefore, even CQL can benefit from DyDiff in these tasks.
>
> [1] Jackson, Matthew Thomas, et al. "Policy-guided diffusion." arXiv preprint arXiv:2404.06356 (2024).
>
> [2] Agarwal, Rishabh, et al. "Deep reinforcement learning at the edge of the statistical precipice." NeurIPS 2021.

---

> > ### Comment · Reviewer_3Lt8 · 2024-11-24
> >
> > Thank you for the detailed reply. While I believe the current revision resolves most of the clarity issues, the reported probabilities of improvement, ranging from 0.52 to 0.6, are not convincing enough to warrant a more favorable evaluation as a "good paper." Therefore, I will maintain my initial evaluation of 6 (weak accept).

---

### Official Review · Reviewer_bgMC · 2024-11-01

**Soundness:** 2
**Presentation:** 3
**Contribution:** 2
**Rating:** 3
**Confidence:** 3

**Summary:**

To address the mismatch between the behavior policy and the learning policy, and to alleviate compounding errors induced by single-step dynamics models, in this paper, a method, namely DyDiff is proposed that can synthesize long-horizon on-policy rollouts for offline policy training, which claims to combine the advantages of both the rollout consistency of single-step dynamics models with arbitrary policies and the long-horizon generation of Diffusion models with less compounding error.

**Strengths:**

The paper is well-motivated and aims to address some important concerns in the literature, the ideas presented in this paper are novel, and both experiments conducted on D4RL and theoretical analysis are provided to show the effectiveness and advantages of the proposed approach.

**Weaknesses:**

While the whole framework is based on the claim that DMs have superior modeling capabilities compared with single-step dynamics models, no theoretical guarantees are provided for such claims, e.g., based on this claim, the conclusion of Theorem 1 shows that \epsilon_m \approx \epsilon_d, however, in the one hand, no theoretical are provided, in the other hand, in some cases, this may be not true. Also, it is unclear why just replacing the generated actions with the action of the learning policy, the mismatch issue can be addressed as the learning policy is also obtained from the real training dataset.

**Questions:**

1. In theorem 1, the authors claimed that over long horizons, the single-step error bound \epsilon_m is even less than the accumulative multi-step error bound \epsilon_d as DMs has superior modeling capabilities, and then concluded that DMs are better for long-horizon rollout than single-step dynamics model. However, no theoretical guarantees are provided.
2. In the proposed approach, states and actions are first generated simultaneously, then the generated actions are replaced with the real ones. Compared with DecisonDiffuser where only state sequences are generated, what are the advantages of the proposed method?
3. In which conditions, can Assumptions 1 and 2 be satisfied?
4. In the experimental results, only the training time of DyDiff is reported. The training time of other baselines should also be stated and compared with the training time of DyDiff.

---

> ### Author Response · Authors · 2024-11-21
>
> - Q1: Also, it is unclear why just replacing the generated actions with the action of the learning policy, the mismatch issue can be addressed as the learning policy is also obtained from the real training dataset.
> - A1: We would like to first clarify our motivation and the policy mismatching problem. This problem arises when the synthetic data do not follow the distribution induced by the learning policy. Although the learning policy is also obtained from the real dataset, it is continuously updated, and its induced distribution keeps changing. By using the actions of the learning policy, we explicitly enforce the generated trajectory distribution to stay closer to the changing distribution in each iteration. In our motivating example in Figure 1, we have shown that augmenting data following the same distribution to the policy will promote more on policy learning, which is further supported by the improvements in our main experiments.
>
> - Q2: In theorem 1, the authors claimed that over long horizons, the single-step error bound \epsilon_m is even less than the accumulative multi-step error bound \epsilon_d as DMs has superior modeling capabilities, and then concluded that DMs are better for long-horizon rollout than single-step dynamics model. However, no theoretical guarantees are provided.
> - A2: We apologize that our statements may not be so clear that misunderstanding arises. Both $\epsilon_m$ and $\epsilon_d$ are conditions assumed to be already known. Because they model the errors related to complicated neural networks that can hardly be analyzed theoretically, we can only evaluate their emperical values. In fact, our theoretical analysis follows a common approach in model-based reinforcement learning used in many previous works [1,2,3], and they only provide empirical error bounds as well. In the uploaded revision, we have cleared our statement in Section 4.3, marked in red.
>
> - Q3: In the proposed approach, states and actions are first generated simultaneously, then the generated actions are replaced with the real ones. Compared with DecisonDiffuser where only state sequences are generated, what are the advantages of the proposed method?
> - A3: Thanks for your valuable question. We motivate differently from DecisionDiffuser in that we aim to build a dynamics model allowing interaction with other policies. In contrast, DecisionDiffuser binds the dynamics model and the policy, so it cannot interact with arbitrary policies. It fills the actions via a pre-trained inverse dynamics model, which does not introduce any new information to the generated sequence. As for DyDiff, the pre-trained diffusion model acts like a dynamics model. Though it first generates states and actions simultaneously, the action sequence is then replaced by actions given by the learning policy itself. Such replacement is the interaction between the diffusion model and the policy, outputing synthetic data that follow both the dynamics and the distribution induced by the policy.
>
> - Q4: In which conditions, can Assumptions 1 and 2 be satisfied?
> - A4: We apologize for the confusion caused by our unclear explanation. In fact, Assumption 1 can be decomposed into the following two assumtions:
>
>   - Assumption 1.1: $\max_t D_\mathrm{TV}(T_d(s_t|s_0,\tau_a) \| T(s_t|s_0, \tau_a)) \le \epsilon_{s,d}$
>
>   - Assumption 1.2: $\max_t D_\mathrm{TV}(T_d(s_t|s_0, \tau_{a,d}) \| T_d(s_t|s_0, \tau_{a})) \le C_{a,d}\max_t \|\tau_{a,d} - \tau_a\|$
>
>   Using the triangular inequality of $D_\mathrm{TV}$, we can obtain Assumption 1. Now, assumption 1.1 is the same as the condition in Theorem 1.
>
>   For Assumptions 1.2 and 2, they are similar to the Lipschitz condition. Assumption 1.2 bounds the change in the state distribution when the action sequence changes, and Assumption 2 bounds the change in the action distribution when the state changes. In practice, when input states and actions do not fall far from the data coverage of the training set, these assumptions can be assumed to hold. In the far-out-of-distribution region, the accuracy of models becomes too low for us to predict their behavior.
>   In the uploaded revision, we have added this explanation and the detailed derivation in Appendix D.4.
>
> - Q5: In the experimental results, only the training time of DyDiff is reported. The training time of other baselines should also be stated and compared with the training time of DyDiff.
> - A5: Thanks for your advice. We have updated the training time of other baselines in Appendix E.7 and included this as a limitation in Section 6. Also, we would like to point out that the training time in offline RL is usually less important than that in online RL.
>
> [1] Janner, Michael, et al. "When to trust your model: Model-based policy optimization." NeurIPS 2019.
>
> [2] Lai, Hang, et al. "Bidirectional model-based policy optimization." ICML, 2020.
>
> [3] Yu, Tianhe, et al. "Mopo: Model-based offline policy optimization." NeurIPS 2020.

---

> > ### Comment · Reviewer_bgMC · 2024-11-27
> >
> > Thanks for the authors' response. The response has addressed some of my concerns. However, for Q2, the authors stated that they can only evaluate the empirical values of  \epsilon_m and epsilon_d. In such a case, how can we claim that \epsilon_m is less than \epsilon_d?  If this assumption cannot be satisfied, does Dydiff still have theoretical advantages over single-step models?
> > One more question:
> > In Dydiff, high-reward trajectories are selected to improve the quality of the training data by pre-training a reward model. Is such a filtering mechanism also adopted for other data synthesizers ( e.g., SynthER)  in the experiments? That only the high-quality synthesis data is used for the data augmentation.

---

> ### Author Response · Authors · 2024-11-27
>
> Thank you for your careful review and insightful feedback on our paper. We have made every effort to address your comments, including:
>
> - Our motivation for replacing the generated actions with the actions derived from the learning policy.
> - A detailed explanation of the theoretical analysis included in the revised paper.
> - The key differences between DyDiff and DecisionDiffuser.
> - The training times of other baselines.
>
> If you have any further questions or concerns regarding our paper or responses, please do not hesitate to let us know. We look forward to engaging in constructive discussions to further improve our paper.

---

> ### Author Response · Authors · 2024-11-28
>
> - Q1: In such a case, how can we claim that \epsilon_m is less than \epsilon_d? If this assumption cannot be satisfied, does Dydiff still have theoretical advantages over single-step models?
> - A1: We would like to clarify the purpose of our theoretical analysis. The analysis aims to demonstrate that if
> $$
> \frac{\gamma}{1-\gamma} \frac{\epsilon_m}{\epsilon_d} > 1 \quad (*),
> $$
>
> using non-autoregressive generative models, such as Diffusion Models, is preferable to single-step models. In our experiments, we find that condition $(∗)$ holds for the Diffusion Model employed in DyDiff, which we believe benefits from generating all timesteps simultaneously, thereby avoiding compounding errors.
>   However, the condition $(\*)$ can only be determined emperically. For example, consider a Diffusion Model without training—it would generate random samples for which $(\*)$ does not hold, and it would clearly not outperform single-step models. Thus, we do not claim that Diffusion Models inherently have theoretical advantages over single-step models. Instead, under empirical conditions, they are very likely to outperform single-step models.
>
> - Q2: In Dydiff, high-reward trajectories are selected to improve the quality of the training data by pre-training a reward model. Is such a filtering mechanism also adopted for other data synthesizers ( e.g., SynthER) in the experiments? That only the high-quality synthesis data is used for the data augmentation.
> - A2: Thank you for your question, and we apologize for the lack of detailed analysis regarding our filtering mechanism. Although the original SynthER does not include a filtering scheme, following your suggestion, we applied the softmax filter to the data generated by SynthER, sampling transitions based on the softmax of their rewards. We then trained TD3BC agents on the filtered datasets, with the results provided in Appendix E.2 of our revised paper. The results indicate that directly picking out high-reward transitions leads to a slight improvement in performance compared to the original SynthER. However, the performance gains are primarily observed in relatively simple tasks, such as Hopper and Walker2d. Conversely, filtered SynthER underperforms relative to the original SynthER on more complex tasks like HalfCheetah and Maze2d-large. This may occur because selecting high-reward transitions limits the training data to better but less accessible regions, which does not necessarily benefit policy learning. For DyDiff, we apply the filtering scheme at the trajectory level, preserving the complete paths leading to high-reward regions.
>
> Thank you again for raising such valuable questions, which have helped us further improve the paper. We hope the above responses address your concerns. If they do, we would greatly appreciate it if you could consider increasing the paper's rating.

---

> > ### Comment · Reviewer_bgMC · 2024-11-28
> >
> > One of the three claimed main contributions of the paper is that theoretical analysis for non-autoregressive generation can be provided, that, compared to single-step models, DyDiff tightens the return gap between executing the policy in the real and the learned dynamics by a substantial factor much greater than 1, which can only hold in the case that  \epsilon_m is no less than \epsilon_d (sorry for the typos in the initial comment ).
> >
> > Note that  \epsilon_m is the singe-step error and \epsilon_d is the accumulative multi-step error, even DMs has superior modeling capabilities, in many cases, this assumption can not be satisfied, which means that we cannot prove DyDiff can tighten the return gap and Theorem 1  and its subsequent conclusion are meaningless.
> >
> > Without Theorem 1 and its subsequent conclusion, in my opinion, the paper's contribution is quite limited. The main contribution is using the actions of the learning policy to enforce the generated trajectory distribution to stay closer to the changing distribution in each iteration, which also lacks theoretical support.

---

> > > ### Author Response · Authors · 2024-11-28
> > >
> > > Thank you for your valuable discussion. We apologize for not describing the meaning of Theorem 1 clear enough, and would like to clarify two points to address your concerns:
> > >
> > > 1. The corollary of Theorem 1, which states that non-autoregressive models tighten the return gap, holds if $\boldsymbol{\frac{\gamma}{1-\gamma}} \cdot \frac{\epsilon_d}{\epsilon_m} > 1$. In typical RL tasks, $\gamma$ is often set to 0.99. Thus, this condition **holds if $\epsilon_d > \epsilon_m / 99$**, which is relatively easy to satisfy given the superior modeling capabilities of DMs. We have also provided empirical results to demonstrate that this condition is met in DyDiff.
> > >
> > > 1. Theorem 1 offers **a general analysis of the generative ability of non-autoregressive models**, not just DyDiff. The purpose of this theory is not only to explain why DyDiff works but also to encourage the broader adoption of non-autoregressive models for trajectory generation. We have revised the third contribution in the introduction and Line 341 in Section 4.3 to clarify that Theorem 1 applies to general non-autoregressive models, with DyDiff being just one application.
> > >
> > > We thank you once again for prompting us to refine our paper and look forward to further discussions.

---

> > > > ### Comment · Reviewer_bgMC · 2024-11-29
> > > >
> > > > Thanks for the authors’ response and efforts to improve the quality of the paper. However, as mentioned in the previous reviews, in my opinion, the paper’s contribution is quite limited without Theorem 1 and its subsequent conclusion. However, the conclusion drawn from Theorem 1 relies heavily on the accuracy of DMs. Although the authors have emphasized that they have provided empirical results to demonstrate that this condition can be met (single-step error \epsilon_m is no less than the accumulative multi-step error \epsilon_d, a simple experiment is conducted to compute the MSE of rollouts generated by both models), but only such a simple experiment is obviously not enough, more experimental results can be provided to show that \epsilon_d can be greater than \epsilon_m.
> > > >
> > > > In the response, the authors pointed out that \lambda is often set to 0.99, please give the reference. Even \lambda is usually set to 0.99, it does not mean the conclusion drawn from Theorem 1 is theoretically valid. Without Theorem 1 and its subsequent conclusion, in my opinion, the main contribution of the paper is that the authors experimentally demonstrated in several domains that with the superior modeling capabilities of DMs techniques, by using the actions of the learning policy to replace the actions of generated trajectory, the distribution can stay closer to the changing distribution in each iteration (which also seems lack of theoretical support), better performance can be achieved, which in my view, cannot meet the bar of ICLR. Also, taking the training time and model size of DyDiff into account, which is much higher than that of SynthER, I am still leaning to reject this paper, but would not be upset if it is accepted.

---

> ### Author Response · Authors · 2024-11-29
>
> Thank you for your question regarding Theorem 1, which is now the key disagreement between us. We would like to further clarify our response as follows:
>
> - We sincerely apologize for any confusion caused by the typo in our previous response. The factor should be expressed as $\frac{\gamma}{1-\gamma} \cdot \frac{\epsilon_m}{\epsilon_d}$, consistent with the notation in the paper. This value is greater than $1$ if $\epsilon_d < \frac{\gamma}{1-\gamma} \epsilon_m$, and specifically, $\epsilon_d < 99 \epsilon_m$ when $\gamma=0.99$. In other words, a DM achieves a better return gap bound than a single-step model as long as its error is less than $\frac{\gamma}{1-\gamma}$ times that of the single-step model. With this corrected formulation, this condition aligns with intuition and much easier to satisfy compared to the incorrect version, especially given the superior modeling capabilities of DMs.
>
> - For the value of $\gamma$, we provide a partial list of commonly used RL algorithms, including both online and offline RL methods, where $\gamma=0.99$ is a standard setting.
>   - Online RL:
>     - DQN[1], Extended Data Table 1
>     - DDPG[2], Section 7
>     - TRPO[3], Table 2, 3
>     - PPO[4], Table 3, 4, 5
>     - SAC[5], Table 1
>     - TD3[6], Table 3
>     - MBPO[7], in its official implementation
>   - Offline RL:
>     - CQL[9], Section F
>     - IQL[10], in its official impelementation
>     - TD3BC[11], Table 4
>     - MOPO[12], in its official impelementation
>     - DiffQL[13], in its official impelementation
>
>   Additionally, most RL algorithms set $\gamma > 0.9$, with very few works using $\gamma < 0.5$. This is because $\gamma < 0.5$ results in highly myopic policies, where long-horizon rollouts are unnecessary.
>
> - Regarding the theoretical validity of Theorem 1, we would like to clarify that incorporating parameters requiring empirical determination is a common practice in theoretical analyses of MBRL[7][12][14]. For instance, in the widely used MBPO[7], Theorem 4.3 includes $\epsilon_\pi$ and $\epsilon_{m'}$, which are left for empirical determination. As our paper adopts a similar framework to MBPO, such parameters also need to be empirically determined.
>
>   In MBPO, the theoretical analysis concludes with the following statement:
>
>   > While this insight does not immediately suggest an algorithm design by itself, we can build on this idea to develop a method that makes limited use of truncated, but nonzero-length, model rollouts.
>
>   Following this reasoning, we do not believe the inclusion of empirical parameters undermines the validity of Theorem 1. Instead, our theoretical analysis is intended to provide insights into why DMs are advantageous, rather than to offer a definitive guarantee that DMs will always outperform other approaches.
>
> [1] Mnih, Volodymyr, et al. "Human-level control through deep reinforcement learning." nature 518.7540 (2015): 529-533.
>
> [2] Lillicrap, T. P. "Continuous control with deep reinforcement learning." arXiv preprint arXiv:1509.02971 (2015).
>
> [3] Schulman, John. "Trust Region Policy Optimization." arXiv preprint arXiv:1502.05477 (2015).
>
> [4] Schulman, John, et al. "Proximal policy optimization algorithms." arXiv preprint arXiv:1707.06347 (2017).
>
> [5] Haarnoja, Tuomas, et al. "Soft actor-critic: Off-policy maximum entropy deep reinforcement learning with a stochastic actor." International conference on machine learning. PMLR, 2018.
>
> [6] Fujimoto, Scott, Herke Hoof, and David Meger. "Addressing function approximation error in actor-critic methods." International conference on machine learning. PMLR, 2018.
>
> [7] Janner, Michael, et al. "When to trust your model: Model-based policy optimization." Advances in neural information processing systems 32 (2019).
>
> [8] Hafner, Danijar, et al. "Mastering diverse domains through world models." arXiv preprint arXiv:2301.04104 (2023).
>
> [9] Kumar, Aviral, et al. "Conservative q-learning for offline reinforcement learning." Advances in Neural Information Processing Systems 33 (2020): 1179-1191.
>
> [10] Kostrikov, Ilya, Ashvin Nair, and Sergey Levine. "Offline reinforcement learning with implicit q-learning." arXiv preprint arXiv:2110.06169 (2021).
>
> [11] Fujimoto, Scott, and Shixiang Shane Gu. "A minimalist approach to offline reinforcement learning." Advances in neural information processing systems 34 (2021): 20132-20145.
>
> [12] Yu, Tianhe, et al. "Mopo: Model-based offline policy optimization." Advances in Neural Information Processing Systems 33 (2020): 14129-14142.
>
> [13] Wang, Zhendong, Jonathan J. Hunt, and Mingyuan Zhou. "Diffusion policies as an expressive policy class for offline reinforcement learning." arXiv preprint arXiv:2208.06193 (2022).
>
> [14] Lai, Hang, et al. "Bidirectional model-based policy optimization." International Conference on Machine Learning. PMLR, 2020.

---

> ### Author Response · Authors · 2024-12-03
>
> Dear Reviewer bgMC,
>
> Thank you for your effort and time on reviewing our manuscript. As today marks the final day of the review period, we would like to kindly ask if there are any remaining questions or concerns that we can address to assist in your evaluation. We greatly appreciate your time and feedback and remain available to provide any clarifications you may need.
>
> Best regards,
>
> Authors of Submission5681

---

### Official Review · Reviewer_U2eQ · 2024-11-04

**Soundness:** 4
**Presentation:** 3
**Contribution:** 3
**Rating:** 6
**Confidence:** 5

**Summary:**

This paper introduces Dynamics Diffusion (DyDiff), a novel method that leverages diffusion models for synthesizing long-horizon, policy-aligned trajectories in offline reinforcement learning. DyDiff addresses the challenge of policy mismatch by iteratively refining generated trajectories to match the learning policy, thereby enhancing long-term rollout accuracy without relying on single-step dynamics models prone to compounding errors. Through theoretical analysis and experiments on D4RL benchmarks, DyDiff demonstrates its effectiveness in improving policy training across various tasks.

**Strengths:**

1. The idea of this paper is very novel and interesting.
2. The writing is clear enough and easy to follow.
3. The theoretical proof of this paper looks precise and interesting.
4. The experimental results of this paper is solid.

**Weaknesses:**

1. The ablation study result in Fig 3.a looks strange to me. If the core idea is to apply learning policy with the learned dynamics by diffusion models, more iteration times should lead to better performance intuitively. The author's explanation is that more iterations will lead to a higher probability of falling out of the data distribution. This is not so convincing to me. The initial trajectory is generated by the learning policy and the inaccurate dynamics (compared with diffusion models). Thus, the later iteration is more like to correct the dynamics with the diffusion model while remaining the policy to be the learning policy. In this case, having more iterations will not increase the risk of falling out of distribution, while it only improves the accuracy of dynamics.

Could the authors provide further analyses and experiments to discuss this problem, including providing a more detailed analysis on how the state distributions and the dynamics consistency change during the iteration?

**Questions:**

1. How do you get the learning policy $\pi$ in the generation process? Does it update iteratively with the learning process? If I understand correctly, does it mean that you first initialize the learning policy then generate trajectories with DyDiff and learn a new policy based on the generated trajectories? Also, what is the frequency of policy updates in relation to the trajectory generation process, and how this will affect the overall performance?

2. Why DyDiff can significantly improve the performance on Maze2d compared with Mojuco? Can the authors provide a more detailed analysis between the results on these two environments?

3. I'm wondering if DyDiff can still improve the performance with some other baselines like Decision Transformer.

4. It's better to include the comparison with other data augmentation methods like MTDiff.

5. How about directly comparing with some diffusion-based planning algorithms using the learning policy as guidance?

---

> ### Author Response · Authors · 2024-11-21
>
> - Q1: Could the authors provide further analyses and experiments to discuss this problem, including providing a more detailed analysis on how the state distributions and the dynamics consistency change during the iteration?
> - A1: Thanks for your question, where we may not describe the refinement process of DyDiff very clearly. We would like to clarify this process again. In the initial trajectory, each action $a_t$ is generated by the learning policy $\pi$ given $s_t$, so the trajectory is consistent with $\pi$. The problem now is $(s_t, a_t, s_{t+1})$ may not follow the real dynamics as the single-step dynamics model is not accurate. After the DM takes $s_0$ and $\tau_a$ as conditions to refine this trajectory, the newly generated $(s_t', a_t, s_{t+1}')$ obeys the real dynamics. However, $a_t$ does not necessarily follow the distribution $\pi(s_t')$. Therefore, the generation process of DM does not remain the policy to be the learning policy, and we further refine this trajectory iteratively with the learning policy and the DM. As an illustrative example, we calculate the total MSE error of the generated trajectorie with a TD3BC policy in the hopper-medium task and plot how it changes with the iteration times in Appendix E.5.
>
> - Q2: How do you get the learning policy $\pi$ in the generation process? Does it update iteratively with the learning process? If I understand correctly, does it mean that you first initialize the learning policy then generate trajectories with DyDiff and learn a new policy based on the generated trajectories? Also, what is the frequency of policy updates in relation to the trajectory generation process, and how this will affect the overall performance?
> - A2: Yes, the learning policy is first initialized and then continuously updated by the underlying offline RL algorithm. In fact, we "update" the policy with both true and generated trajectories rather than learn a "new" policy. As for the frequency of policy updates, lowering this frequency is equivalent to increasing the generated data amount in each generation process. We conduct experiments on hopper-medium-v2 with TD3BC, finding this hardly affects the performance, as listed in the following table, because the proportion of synthetic data used for updating the policy is controlled by the real ratio $\alpha$. We generate 2000 trajectories in the main paper and never tuned this hyperparameter.
>
> | #Synthetic Trajs | 1000 | 1500 | 2000 (Main Paper) | 3000 |
> |---|---|---|---|---|
> | Results | 70.4 $\pm$ 10.6 | 72.1 $\pm$ 6.1 | 71.5 $\pm$ 15.5 | 70.3 $\pm$ 12.4 |
>
> - Q3: Why DyDiff can significantly improve the performance on Maze2d compared with Mojuco? Can the authors provide a more detailed analysis between the results on these two environments?
> - A3: We contribute the significant improvement of DyDiff on Maze2d to the ability of modeling long-horizon distributions, so DyDiff can synthesize high-quality long trajectories. We visualize the synthetic trajectories in Maze2d, as shown in Appendix E.6. In Maze2d tasks, the agent does not receive any reward until it reaches the goal, so effective exploration will accelerate the learning process and lead the learning policy to find a shorter path to the goal. On the contrary, MuJoCo tasks provide dense rewards, so the benefit from exploration is smaller than on Maze2d tasks.
>
> - Q4: I'm wondering if DyDiff can still improve the performance with some other baselines like Decision Transformer.
> - A4: Thanks for your advice about applying DyDiff on Decision Transformer (DT). However, the structure of DyDiff must be modified to be compatible with the return-to-go used in DT, and training DT costs a lot of time. Our experiments are in running, and we will report the results as soon as possible.
>
> - Q5: It's better to include the comparison with other data augmentation methods like MTDiff.
> - A5: Thanks for your advice on comparing DyDiff with MTDiff. We use the official implementation of MTDiff and replace its task list by a single task. We first train the MTDiff-s model on the given dataset and then generate 1M $(s,a,r,s')$ samples as the synthetic dataset $\mathcal{D} _ {\mathrm{MTDiff}}$ . Finally, we train the RL policy on the mixture of $\mathcal{D} _ \mathrm{real}$ and $\mathcal{D}_\mathrm{MTDiff}$. The results are reported in Table 6, Appendix E.4. We find that MTDiff can hardly improve the performance of the underlying policy since it is originally designed for multi-task situations. Its generalization ability from learning knowledge across multiple datasets is significantly weakened in the single-task setting.

---

> ### Author Response · Authors · 2024-11-21
> **Rebuttal Cont.**
>
> - Q6: How about directly comparing with some diffusion-based planning algorithms using the learning policy as guidance?
> - A6: It is hard to directly compare DyDiff with diffusion-based planning algorithms as they do not possess a decoupled policy. Specifically, diffusion-based planning methods, or diffusion planners, usually directly plan long-horizon state-action trajectories, or state trajectories but fill the actions by an inverse dynamics model. In both cases, they determine the action to be executed by the planned trajectories without the help of independent policies. Therefore, they do not use the learning policy as guidance, which is different from DyDiff in principle.

---

> ### Comment · Reviewer_U2eQ · 2024-11-27
> **Official Comment by Reviewer U2eQ**
>
> Thank you very much for addressing the clarification questions raised. I will maintain the score of weak acceptance.

---

### Author Response · Authors · 2024-11-21
**General Response**

We greatly appreciate all reviewers for your careful reviewing and valuable advice. Based on your suggestions, we have revised our paper, including:
- Revising the possibly confusing introduction to the framework of DyDiff in the caption of Figure 2, Line 182, and Line 232-239.
- Clearing the notations from Line 274, Section 4.2.
- Explanation of why $\epsilon_d$ and $\epsilon_m$ cannot be further decomposed in Line 336, Section 4.3.
- The online RL settings in Section 6 now include a time-consuming limitation.
- More details for preliminaries in Appendix B.
- Explanation of assumptions that explain why we make such assumptions and when they are likely to be satisfied, in Appendix D.4.
- Comparison to MTDiff-s in Appendix E.4.
- Explanation and empirical example of how the error of synthetic trajectories changes over the iteration times $M$, in Appendix E.5.
- Visualization of synthetic trajectories on Maze2D tasks in Appendix E.6.
- Training time and model sizes of DyDiff and SynthER in Appendix E.7.

We mark the modified text or the captions of added sections in red.

---

### Author Response · Authors · 2024-11-24
**The discussion will end in 4 days**

Dear Reviewers,

We sincerely thank you for taking the time to review our manuscript and provide valuable feedback. Your insights have been instrumental in improving our work.

As the discussion deadline is now only 4 days away, please let us know if we missed anything or you still have confusion. We greatly appreciate your efforts and look forward to receiving your feedback. Any questions to our paper and rebuttal are always welcomed.

Thank you again for your time and support.

Best regards,

Authors of Submission5681

---

### Author Response · Authors · 2024-12-04
**Rebuttal Summary**

We would like to sincerely thank the reviewers for their thorough and insightful feedback on our paper. We are pleased to know that our responses have helped clarify some of the concerns and confusions regarding our work. In this paper, we introduce **Dynamics Diffusion (DyDiff)**, a novel approach for synthesizing long-horizon rollouts that are both accurate in terms of dynamics and consistent with the learning policy. We demonstrate the effectiveness of DyDiff both theoretically and empirically, with the theoretical analysis highlighting the potential of general non-autoregressive models in long-horizon generation.

Based on the reviewers’ feedback, our paper's strengths include:
- All reviewers agree that the idea is novel, and Reviewers U2eQ, 3Lt8, and kq5G also note that it is well-motivated.
- Reviewers U2eQ, 3Lt8, and kq5G find the paper well-written and easy to follow.
- Reviewers U2eQ and kq5G acknowledge that the theoretical analysis is precise and supports the method and experimental results.

For the weaknesses raised, we have revised the paper as follows:
- Clarified confusing notations, captions, and explanations of the theory.
- Provided additional background knowledge to support the context of our work.
- Expanded the empirical analysis, including a comparison of DyDiff with a new baseline, MTDiff-s, as well as the visualization of synthetic trajectories.

We would like to thank the reviewers once again for their invaluable constructive comments and suggestions. Their feedback has greatly helped us improve both the quality and clarity of our work. We believe that most of the concerns have been addressed through our responses and the revisions made to the paper. We truly appreciate the time and effort the reviewers have dedicated to evaluating our submission.

---

### Meta-Review · Area_Chair_sM1i · 2024-12-22

**Metareview:**

This paper attempts to address the problem in the use of diffusion models for offline RL where the generated trajectories don't align with the learned policy. The approach works by first training a single-step transition model and generating roll-outs using this model and the policy. Next, these trajectories are refined by sampling iteratively from the diffusion model and policy. Finally, generated trajectories are filtered and high-reward ones are stored in a synthetic data which can be used for offline training.

Strengths:
- Reviewers found the approach novel
- Generating trajectory data that is consistent with the policy is desirable and the proposed approach makes sense
- Theoretical guarantees showed that the non-autoregressive approach works better when $\frac{\gamma}{1-\gamma} \frac{\epsilon_d}{\epsilon_m} > 1$, where $\epsilon_m$ is the TV error of the 1-step transition model against the true model, and $\epsilon_d$ is the same for the non-autoregressive model.

Weakness:
- The chief concern is gains are minor
- Given approach is computationally expensive

The main concern here is whether the gains justify the approach. This is raised both by reviewer 3Lt8 and reviewer kq5G. Further, the proposed approach is much more complex and computationally expensive. Training diffusion models is challenging. For these reasons, while I do see a promise here and an important problem, I am recommending weak rejection. I suggest authors improve their approach to get bigger gains, or simplify it significantly where those gains make sense.

**Additional Comments On Reviewer Discussion:**

Reviewers raised the following main concerns:

1. The chief concern is gains are minor. This concern was raised by reviewer 3Lt8 and reviewer kq5G. Further, the approach is computationally expensive. In response, authors toned down their claim.

2. Another concern was raised by reviewer bgMC on the question of how to prove any relation between $\epsilon_d$ and $\epsilon_m$.
One can use statistical learning theory bounds to get bounds on the total variation. E.g., optimizing log-likelihood on $\frac{1}{n} \sum_{i=1}^n \log p_\theta(y_i | x_i)$ where $x_i \sim D$ and $y_i \sim p^\star(\cdot \mid x)$, gives the following bound: $E_{x \sim D}[ || p^\star(Y \mid x) - p_{\hat{\theta}}(Y \mid x) ||_{TV}] \le \sqrt{\frac{2}{n} \log \frac{|\Theta|}{\delta}}$, with probability at least $1-\delta$.

However, even then you are comparing upper bounds with upper bounds. Overall, this concern doesn't bother me and I would be a lot more convinced by the experimental approach that the authors advocate. Further, authors show that for $\gamma=0.99$ which is indeed a common choice in RL, they only require $\epsilon_d \ge \frac{\epsilon_m}{99}$, meaning that $\epsilon_d$ has to satisfy a weaker condition.

3. There were also questions on why more iteration does not lead to better results as one would expect. Reviewer U2eQ raised this and was not convinced by the explanation in the paper. Reviewer kq5G also raised this sensitivity question. Authors argued that generally, 1-2 iterations are enough.

There were other clarity-related questions that the authors addressed reasonably well. Overall, my main concerns remained weak gains at the cost of much higher compute.

---

### Decision · Program_Chairs · 2025-01-22

Reject